# A trait-based root acquisition-defence-decomposition framework in angiosperm tree species

Jiajia Zheng[1,2,3], Grégoire T. Freschet [4], Leho Tedersoo [5,6], Shenggong Li[1,3], Han Yan[7,8], Lei Jiang[9], Huimin Wang[1,2,3], Ning Ma[1,3], Xiaoqin Dai[1,2], Xiaoli Fu[1,2,3] & Liang Kou [1,2,3] ✉

To adapt to the complex belowground environment, plants make trade-offs between root resource acquisition and defence ability. This includes forming partnerships with different types of root associating microorganisms, such as arbuscular mycorrhizal and ectomycorrhizal fungi. These trade-offs, by mediating root chemistry, exert legacy effects on nutrient release during decomposition, which may, in turn, affect the ability of new roots to re-acquire resources, thereby generating a feedback loop. However, the linkages at the basis of this potential feedback loop remain largely unquantified. Here, we propose a trait-based root 'acquisition-defence-decomposition' conceptual framework and test the strength of relevant linkages across 90 angiosperm tree species. We show that, at the plant species level, the root-fungal symbiosis gradient within the root economics space, root chemical defence (condensed tannins), and root decomposition rate are closely linked, providing support to this framework. Beyond the dichotomy between arbuscular mycorrhizal-dominated versus ectomycorrhizal-dominated systems, we suggest a continuous shift in feedback loops, from 'high arbuscular mycorrhizal symbiosis-low defence-fast decomposition-inorganic nutrition' by evolutionarily ancient taxa to 'high ectomycorrhizal symbiosis-high defence-slow decomposition-organic nutrition' by more modern taxa. This 'acquisition-defence-decomposition' framework provides a foundation for testable hypotheses on multi-dimensional linkages between species' belowground strategies and ecosystem nutrient cycling in an evolutionary context.

Since the early Devonian, the progressive emergence of root organs has had profound consequences for land colonisation and diversification of plants[1–3]. During evolution, plants have balanced costs and benefits for better adaptation to complex and ever-changing environments. Sessile growth may have led plants to evolve particularly complex and diverse belowground strategies. When investing carbon (C) belowground, for example, plants may build a root with trait syndromes that range from the very efficient acquisition of soil resources on the one side to effective defences against biotic and abiotic threats for their survival and prosperity on the other side[4]. They can also rely on microbial partners to acquire resources or benefit from protection[5–7]. Such trade-offs between different types of ecological strategies have consequences for the fate of allocated C and nutrients after organs have senesced and died[8–10]. Roots decompose where they grow and die[11], implying that plants may access and preempt the nutrients released by their root litter. Therefore, the trade-off in plant

resource allocation to belowground resource acquisition versus defence[12] has legacy effects on root decomposition, which may further feedback to nutrient reutilization by plants[13,14]. By taking a holistic view of these intrinsic linkages, we propose a feedback loop of 'acquisition-defence-decomposition' (hereafter abbreviated as ADD) operating on root systems driven by the multidimensional belowground adaptation strategies of species (Fig. 1), which may facilitate a comprehensive understanding of plant belowground strategies, species coexistence, and diversity maintenance.

The ADD conceptual framework is essentially determined by the relationships among root traits, which are largely influenced by root-fungal symbiosis[15]. The vast majority of land plants form symbiotic associations with mycorrhizal fungi[16,17]. Plants allocate photosynthate to roots or fungal partners to acquire edaphic resources[6], and to different types of mycorrhizal partners (e.g., arbuscular mycorrhizal (AM) and ectomycorrhizal (EcM)), which results in varying root traits that represent the type of association and their intensity[18]. Mounting evidence has shown strong evolutionary controls over root traits related to mycorrhizal symbiosis, such as diameter[19–21], cortical tissue[22], and branching[20]. Furthermore, condensed tannins, an important root trait related to chemical defence[23], are less abundant in the early diverging magnoliids than in modern rosids[24]. These findings imply possible covariations among root traits associated with 'acquisition' and 'defence' as taxa evolve, yet it remains unclear whether these evolutionary patterns exert influences on the 'acquisition-defence-decomposition' linkages within the ADD conceptual framework.

The two most common mycorrhizal associations in woody plants, AM and EcM, have distinct evolutionary histories and symbiotic pathways (intracellular vs. intercellular)[15,25], leading to large differences in their root traits[18]. Belowground resource acquisition strategies are reflected by the trait-based root economic space (RES), where the root-fungal symbiotic pathway and intensity differentiate AM from EcM plant species[18,26]. In the evolutionarily ancient AM fungal species, hyphae can penetrate cortical cells, forming intracellular arbuscules,

and thus, host plants tend to build roots with a broader cortex space for colonization by AM fungi to acquire limiting soil resources[18]. It has been well documented that thicker AM roots with a greater cortex and higher mycorrhizal dependency are associated with lower concentrations of condensed tannins[24], a phenolic compound slowing down root decomposition[27]. These linkages imply that with the constraint of C, AM tree species relying more on symbiotic fungi may invest less in their belowground chemical defences, thus forming root litter with fewer tannins and faster decomposition.

These patterns in AM plant species may not necessarily hold for more modern EcM species with intercellular symbiosis. EcM plants tend to enhance their collaboration with mycorrhizal fungi by constructing roots with high branching to maximize the area of the symbiotic interface for acquiring limiting soil resources[6,28]. The evolutionary transition from AM to EcM symbiosis implies stronger control of the host plants over symbiotic fungi[16]. For instance, EcM angiosperms develop wall-thickened exodermis with Casparian bands and suberin lamellae and deposit phenolic compounds (e.g., tannins), which function as a physical and chemical barrier to prevent further penetration of symbiotic fungi and pathogens into the inner part of the cortex[29–31]. Therefore, EcM fungi may trigger higher levels of physical and chemical defences in their host plants than AM fungi[32], which may lead to a symbiosis-defence relationship distinct from that in AM species. Furthermore, tannin-rich roots in EcM species could associate with chitin in fungal necromass to form more complex macromolecular polymers that are recalcitrant to decomposers[33,34]. These findings indicate that the intensified chemical defences associated with EcM symbiosis could probably result in slower root decomposition.

The classical mycorrhizal-associated nutrient economy (MANE) hypothesis proposes that AM- and EcM-dominated forests with contrasting nutrient-cycling modes differ in litter quality, which feeds back to plant nutrient uptake[9,25]. Fast-decomposing litters with higher nutrient contents also release nutrients faster and in larger amounts for use by soil organisms and complexation with the soil matrix[35]. Soil microbial

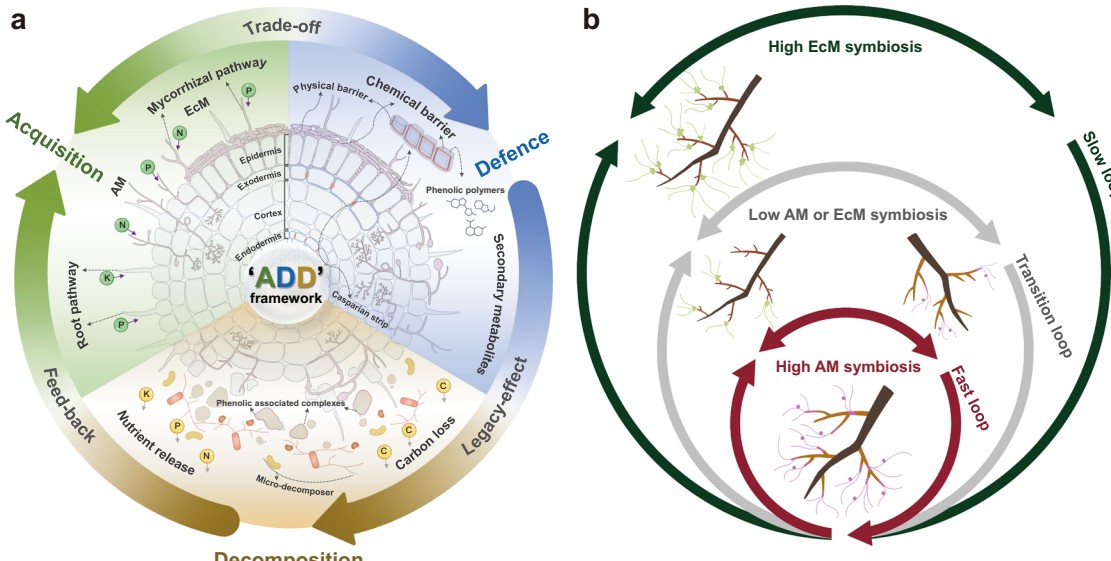

**Fig. 1 | Conceptual framework for the 'acquisition-defence-decomposition (ADD)' loop in the root system. a** To adapt to the complex soil environment, plants either rely on their absorptive roots (i.e., root pathway) or cooperate with symbiotic fungi (arbuscular mycorrhizal or ectomycorrhizal pathways) to acquire soil resources (e.g., nitrogen N; phosphorus P; potassium K) while producing defensive substances (e.g., secondary metabolites) for protection against pathogens and herbivores. There could be a trade-off between 'acquisition' and 'defence' for given photosynthetic carbon, which may exert a legacy effect on carbon (C) loss and nutrient release during decomposition by mediating root chemistry. These

released nutrients can further feedback to nutrient reutilization by root systems considering that absorptive roots decompose where they live and die. We propose that these root processes are intrinsically linked to form an ADD loop. **b** Differences in belowground strategies with respect to mycorrhizal associations (type and intensity) are assumed to drive a continuum of feedback loops from a 'high AM symbiosis-low defence-fast decomposition' loop to a 'high EcM symbiosis-high defence-slow decomposition' loop with several intermediate transition loops. The left diagram is inspired by Heike et al.[79] and Salas-González et al.[80].

communities fed with organic matter inputs of lower C:nutrient ratio further tend to release a larger proportion of nutrients for the use of plants[36]. Such fast-decomposing litter also favours the development of copiotrophic (bacterial-dominated) microbial communities speeding up nutrient cycling in soils, whereas slower-decomposing litter favours oligotrophic (fungal-dominated) communities better adapted to degrade organic compounds with low nutrient content[37]. In this context, EcM tree species, characterized by an organic-nutrient economy, benefit from habitats with slow organic matter decomposition and nutrient release. Conversely, AM tree species thrive in conditions of faster C and nutrient cycling that they promote. However, the MANE framework remains disconnected from current theories in root ecology and aspects of the root economics space[26], belowground litter decomposition and nutrient recycling[38], and is based on a dualistic view of monolithic AM versus EcM mycorrhizal associations. The ADD conceptual framework (Fig. 1) intends to fill in these gaps by specifically focusing on belowground aspects of root economics, particularly the trade-off in root traits, and developing a more quantitative trait-based approach to account for the large variation in trait values across and within tree species mycorrhizal type.

In this study, we focused on absorptive roots (the most distal 1st- and 2nd-order roots within the branching root system) of 90 angiosperm tree species (65 AM and 25 EcM) spanning two temperate and two subtropical forests in China. These species represent a subset of species from Yan et al. [18]. To reveal the strategies for belowground resource acquisition, we characterized the RES of these species by integrating root morphological (root diameter (RD), specific root length (SRL), root tissue density (RTD)), architectural (branching intensity (BI)), anatomical (cortex thickness (CT)), and chemical (nitrogen (N) concentration) traits with the concentration of condensed tannins, which indicate the chemical defence of roots against herbivores[23,24,39]. We also performed a microcosm experiment to quantify the decomposition rate of absorptive roots in these tree species. Our overarching hypothesis is that there are strong linkages and trade-offs between resource acquisition strategy, defence capacity, and decomposition rate in the absorptive root system. Considering the distinct evolutionary histories, symbiotic pathways, and nutrient economies of AM and EcM species[9,25] and large variation in trait values within each mycorrhizal type, we further hypothesized that evolutionary shifts from AM to EcM association mediate the 'acquisition-defence-decomposition' linkages, thus forming a progressive gradient of nutrient-cycling modes. Here, we provide evidence for our 'acquisition-defence-decomposition' conceptual framework by documenting a continuum in root and mycorrhizal traits responsible for multiple feedback loops among plant strategies of nutrient acquisition, defence and afterlife effects on decomposition. This work substantiates the classical MANE hypothesis and, using a trait-based approach expands it beyond the restrictive dichotomy between AM-dominated versus EcM-dominated systems.

## Results

### Root 'acquisition-defence-decomposition' linkages
We observed large interspecific variations in absorptive root traits across 90 angiosperm tree species, with RD and concentrations of tissue N and condensed tannins showing stronger phylogenetic conservation than other root traits (Supplementary Table 1). For tree species ($n = 70$) in the major taxa (only families with at least three species were retained), we found that RD, CT, and concentrations of tissue N and condensed tannins showed significant phylogenetic signal. We further performed a phylogenetic principal components analysis (pPCA) on six key root traits of 90 angiosperm tree species. The first two axes of pPCA accounted for 69.2% of the total trait variation (Supplementary Table 3). The first axis (PC1) correlated with RD, CT and BI, and the second axis (PC2) correlated with SRL and RTD (Fig. 2a, Supplementary Table 3).

To test the validity of the ADD framework, ordinary least squares (OLS) regression was applied to examine the linkage among root economics space (pPCA axis scores), condensed tannins and decomposition rate. Specifically, we found that along the direction of higher RD/CT and lower BI on the PC1, the condensed tannins declined ($R^2 = 0.224$, $P < 0.001$, Fig. 2b), while the root decomposition rate showed an increasing trend ($R^2 = 0.189$, $P < 0.001$, Fig. 2d). Furthermore, when pPCA was performed by including condensed tannins and decomposition rate, we found that these two traits were coaxial with RD, CT and BI. Specifically, RD and CT were positively correlated with decomposition rate and negatively with condensed tannins (Supplementary Fig. 2; Supplementary Table 5). These linkages were also supported by the results of bivariate trait relationships based on both OLS and phylogenetic generalized least squares (PGLS) models. RD was negatively correlated with condensed tannins and positively correlated with root decomposition rate. Condensed tannins were positively correlated with BI and negatively correlated with CT (Supplementary Fig. 3). In addition, the root decomposition rate decreased significantly with increasing condensed tannins (Fig. 2c).

We also used the OLS regression to test how the evolutionary history of plants drives the 'acquisition-defence-decomposition' linkage across major plant families. We found a positive relationship between the RD/CT-BI axis scores and divergence time at the family level ($R^2 = 0.438$, $P = 0.007$, Fig. 2e). A higher RD and CT was associated with evolutionarily ancient taxa (e.g., Magnoliaceae and Lauraceae), while a higher BI became common in recently diverged taxa (e.g., Juglandaceae and Betulaceae). When considering specific root traits, divergence time was positively correlated with RD and CT, but negatively correlated with BI (Supplementary Fig. 7). There were no significant relationships between divergence time and condensed tannins and decomposition rate ($P > 0.050$; Supplementary Fig. 7d, h).

### Mycorrhizal effects on 'acquisition-defence-decomposition' linkages
We found that absorptive root traits differed between mycorrhizal types. RD, CT, root N concentration, and mass loss were significantly higher, and BI was significantly lower in AM than in EcM tree species (Supplementary Table 2). The pPCA representations of AM and EcM tree species subsets were similar to that of all species. Based on the results of pPCA, the first two axes accounted for 72.5% and 59.2% of the total variation for AM species and EcM species, respectively (Supplementary Table 3).

The linkages of pPCA axes with condensed tannins and root decomposition rate differed between mycorrhizal types. In AM species, there was a significant tendency towards decreasing condensed tannins ($R^2 = 0.311$, $P < 0.001$, Fig. 3b) and increasing root decomposition rate ($R^2 = 0.091$, $P = 0.014$, Fig. 3d) with greater RD and CT on the PC1. When condensed tannins and decomposition rate were added to the pPCA of AM species, they aligned very closely to the PC1 axis, with little overall changes in the pPCA representation (Supplementary Fig. 2; Supplementary Table 5). In addition, the results of bivariate trait relationships showed that RD was negatively correlated with condensed tannins but positively correlated with root decomposition rate; condensed tannins were positively correlated with BI and negatively with CT (Supplementary Fig. 4). EcM tree species showed a significant tendency towards increasing condensed tannins with increased BI on the PC2 ($R^2 = 0.258$, $P = 0.010$, Fig. 3f). When condensed tannins and decomposition rate were added to the pPCA of EcM species, they aligned very closely to the PC1 axis, along with RD and BI. Specifically, BI was correlated negatively with decomposition rate and positively with condensed tannins (Supplementary Fig. 2; Supplementary Table 5). Based on results of bivariate trait relationships in EcM species, RD was negatively related to condensed tannins and positively related with root decomposition rate, and BI was positively correlated with condensed tannins (Supplementary Fig. 5). In addition, root

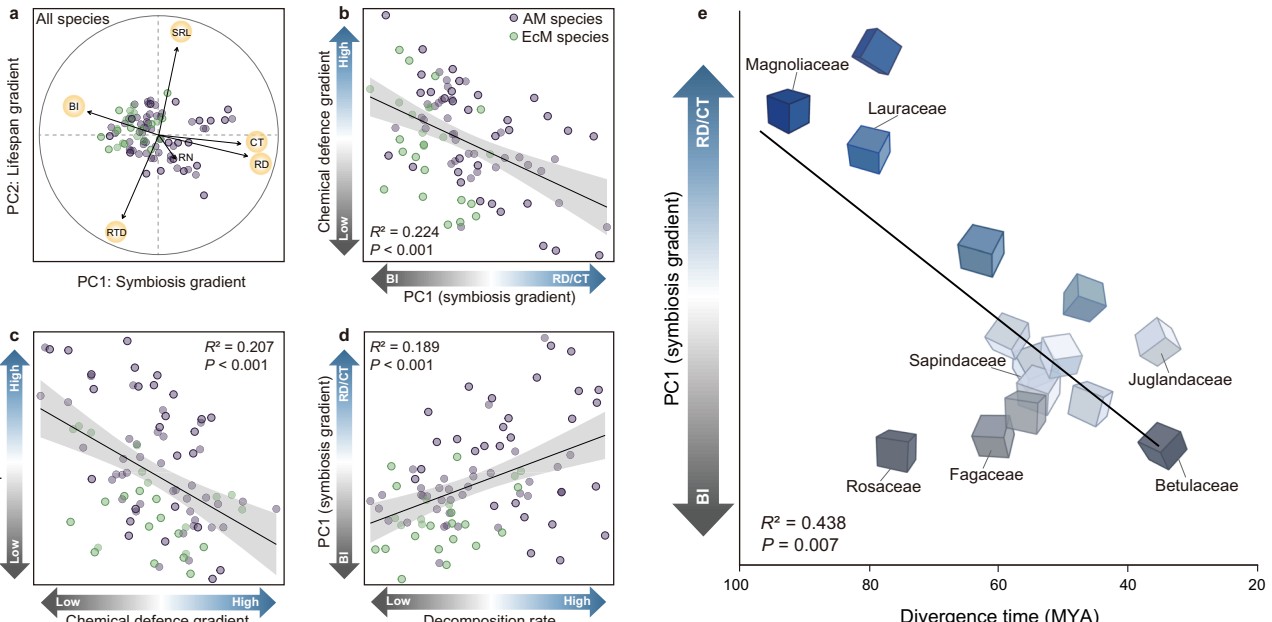

**Fig. 2 | Linkages among the root economics space, chemical defence and decomposition as well as the integrative strategies of species based on the 'acquisition-defence-decomposition' loop.** Root trait biplots of phylogenetic principal component analysis (pPCA) for 90 angiosperm tree species (**a**). The purple and green circles represent arbuscular mycorrhizal ($n = 65$) and ectomy-corrhizal ($n = 25$) tree species, respectively. For more details on pPCA scores, see Supplementary Table 3. Within the root economics space, the 'symbiosis gradient' (the first principal component, PC1) representing the reliance of roots on fungi ranges from the high reliance on EcM fungi with high branching intensity (BI) to the strong reliance on AM fungi with high root diameter (RD) and cortex thickness (CT); and the 'lifespan gradient' (the second principal component, PC2) ranges from the short-lived roots with high specific root length (SRL) to the long-lived roots with high root tissue density (RTD). RN root nitrogen concentration. The linear regressions are fitted among scores of the SRL-RTD axis or RD/CT-BI axis within the root economics space, condensed tannins concentrations, and decomposition rate (**b**, **c**, **d**). The chemical defence gradient refers to the concentrations of condensed tannins from low to high, and the decomposition rate indicates the root mass loss from low to high. The colour from grey to blue represents the gradients of the variables ranging from low to high on the corresponding axis, except that represent the transition of dominant traits (from BI to RD/CT) on the 'symbiosis gradient'. Solid lines indicate regression lines, shaded areas around each line are 95% confidence intervals (*F*-test). The linear regressions are also fitted between the score of RD/CT-BI axis (PC1) and the divergence time in evolutionary history (MYA, million years ago), of major taxonomic groups ($n = 15$) at the family level (only families with at least three species were retained) (**e**). The cubes represent a multidimensional strategy along the 'acquisition-defence-decomposition' continuum driven by the 'symbiosis gradient'. Source data are provided as a Source Data file.

decomposition rate decreased with condensed tannins in both AM ($R^2 = 0.299$, $P < 0.001$, Fig. 3c) and EcM plant species ($R^2 = 0.459$, $P < 0.001$, Fig. 3g).

By using the OLS analysis, we further explored the evolutionary effect on the 'acquisition-defence-decomposition' linkage for the major taxa in AM tree species but not for EcM tree species that have limited number of taxa at the family level. Consistent with the results without distinguishing mycorrhizal types, we found that the divergence time of AM tree species estimated at the family level was positively correlated with the RD/CT-BI axis ($R^2 = 0.393$, $P = 0.039$, Supplementary Fig. 6) but not with condensed tannins and decomposition rate ($P > 0.050$, Supplementary Fig. 8d, h). Furthermore, divergence time was positively correlated with RD ($R^2 = 0.447$, $P = 0.025$; Supplementary Fig. 8a) and CT ($R^2 = 0.406$, $P = 0.035$, Supplementary Fig. 8b) of AM tree species at the family level.

## Discussion

We hypothesized here that several key root processes, including nutrient acquisition, chemical defence, and root decomposition, are linked via trade-offs, which can be characterized by root trait syndromes[18,40,41]. By comparing root traits closely related to these processes across 90 tree species, we found a suite of important belowground trait gradients and linkages. First, root resource acquisition can be reflected by a bi-dimensional RES, consistent with the findings in Yan et al.[18]. The SRL-RTD axis represented a 'lifespan gradient' from acquisitive-strategy species with short root lifespan to conservative-strategy species with long root lifespan, while the RD/CT-

BI axis represented a 'symbiosis gradient' showing the dependency of plant nutrient acquisition on their fungal partners (Fig. 2a). Second, a trade-off between the type of mycorrhizal association (AM vs EcM) and some aspects of chemical defence were indicated by the negative relationship of the score of RD/CT-BI axis (symbiosis gradient) in the RES with the concentration of condensed tannins (Fig. 2b, Supplementary Fig. 2). Our results further suggest that this trade-off may have exerted a legacy effect on root decomposition, as indicated by the negative relationship between concentrations of condensed tannins and root mass loss (Fig. 2c, Supplementary Fig. 2). These trade-offs and legacy effects jointly bring preliminary evidence for the role of tree roots and mycorrhizal type in driving a feedback loop (ADD conceptual framework; Fig. 1).

The RES revealed here differed from the axis representations in Bergmann et al.[26] where RD-SRL axis represents a "collaboration" gradient and RTD-N axis represents a "conservation" gradient. Here, we argue that these two representations of the RES are overall consistent with each other, and the fundamental divergence stems from our sampling of tree species displaying little variation in RD. Based on the theoretical formula: $SRL = 4/(\pi \times RTD \times RD^2)$, any change in RD has much stronger weight on SRL than change in RTD because RD is quadratic to SRL[42]. In contrast to the global dataset of Bergmann et al.[26], who used a broad definition of fine roots, our dataset is restricted to a narrow range of root orders (1st- and 2nd-order roots). It also focuses on a narrower group of plants ('trees' only rather than the 'woody species' category), and lineage (angiosperms). This stringent selection is likely responsible for the narrow interspecific variation in RD, which led to the

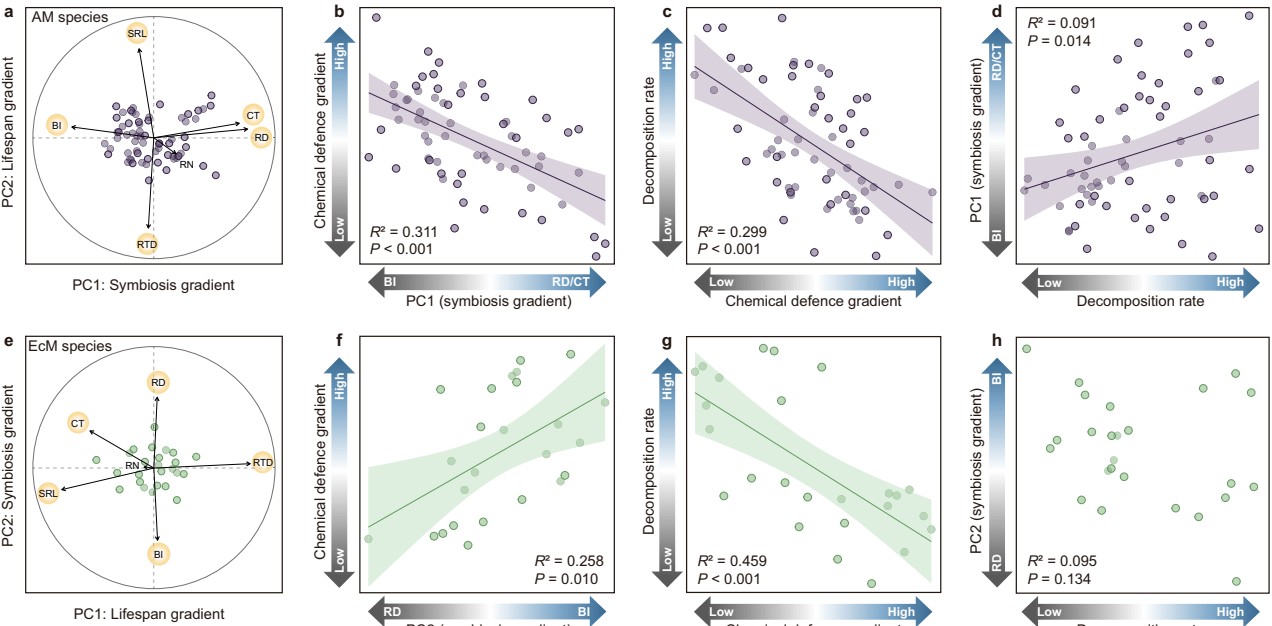

**Fig. 3 | Linkages among the root economics space, chemical defence, and decomposition for arbuscular mycorrhizal (AM) and ectomycorrhizal (EcM) tree species.** Root trait biplots of phylogenetic principal component analyses (pPCAs) for AM (**a**, purple circle) and EcM (**e**, green circle) tree species. For more details on pPCA scores, see Supplementary Table 3. Within the root economics space, the 'lifespan gradient' the first principal component (PC1) for EcM species and the second principal component (PC2) for AM species, ranges from short-lived roots with high specific root length (SRL) to long-lived roots with high root tissue density (RTD); while the 'symbiosis gradient' for AM (PC1) and EcM (PC2) species, representing the reliance of roots on fungi ranges from weak to strong. The strong reliance on fungi is represented by high branching intensity (BI) for EcM species and represented by high root diameter (RD) or cortex thickness (CT) for AM species. RN root nitrogen concentration. The regressions are fitted among scores of the RD/CT-BI axis within the root economics space, chemical defence gradient, and decomposition rate (**b**, **c**, **d**, **f**, **g**, **h**). Except for the transition of dominant traits from BI to RD/CT on 'symbiosis gradient', the gradients of the other variables (e.g., the concentrations of condensed tannins, decomposition rate) ranging from low to high are denoted in colour from grey to blue on the corresponding axis. Solid lines indicate regression lines, shaded areas around each line are 95% confidence intervals (*F*-test). Source data are provided as a Source Data file.

decoupling between RD and SRL. In contrast, we found a large variation in RTD across species leading to a strong coupling between SRL and RTD. Interestingly, our results, based on a strong representation of subtropical forests (60%, Supplementary Data 1), support previous observations that changes in SRL may be more strongly regulated by RTD compared to RD in tropical forests[43]. Nonetheless, we also acknowledge that the low variation in RD observed in our study may not be representative of global tree root variation, which deserves further research.

Tree species position along the 'symbiosis gradient' is associated with their evolutionary history, indicated by the close link between the RD/CT-BI axis scores and divergence time at the family level (Fig. 2e; Supplementary Fig. 6). Along evolutionary lines, AM species may have reduced their dependence on symbiotic fungi, represented here by the decreased RD and CT with divergence time. Meanwhile, following the emergence of EcM associations, EcM species typically show a consistently strong reliance on EcM fungi, as reflected here from high BI[20–22]. Considering the tight linkages among root-fungal symbiosis, chemical defence, and root decomposition, we emphasize that the evolutionary trend of root-fungal symbiosis has major consequences on multidimensional strategies of the root system (Fig. 2e), although we did not find here direct evidence that evolution acts upon chemical defence and root decomposition (Supplementary Fig. 7). Therefore, the root ADD strategies may have evolved from 'high AM symbiosis-low defence-fast decomposition' (e.g., Magnoliaceae) to 'low AM symbiosis-high defence-slow decomposition' (e.g., Sapindaceae) and to 'high EcM symbiosis-high defence-slow decomposition' (e.g., Betulaceae). This strategic transition may result from trade-offs and long-term adaptation to complex climatic and edaphic environments over the evolutionary history of plants. Notably, non-mycorrhizal (NM)

plants that may evolve from facultative AM or NM-AM plants often emerge in ultra-poor soils[44]. To exclude mycorrhizal fungi from their roots, NM plants have evolved potent chemical defence mechanisms by accumulating relatively advanced secondary metabolites (e.g., alkaloids and cyanogens), contrasting with mycorrhizal plants that are more likely to contain primitive secondary metabolites, such as tannins[16,44]. This basic symbiosis-defence difference may result in contrasting root ADD strategies in NM plants compared to mycorrhizal plants, requiring further investigation.

Considering each tree mycorrhizal type separately, the linkages between the 'symbiosis gradient' and chemical defence were distinct (Fig. 3b, f). This finding is largely consistent with previous knowledge of AM versus EcM differences in anatomical and chemical root trait organization[17]. We found negative relationships of the dominant traits (RD and CT, Supplementary Fig. 4) and the RD/CT-BI axis scores with the concentrations of condensed tannins in AM species only (Fig. 3b), which concurs with a recent study reporting greater chemical protection associated with lower mycorrhizal dependency in AM species[24]. AM species with more primitive symbiotic pathways tend to construct thicker roots with a larger cortex to provide habitat for symbiotic AM fungi[18,28]. Compared to the nonmycorrhizal root system, mycorrhizal symbiosis may consume a large amount of C for the construction and maintenance of the symbiont habitat (wide cortex) and for producing the extraradical mycelium[44]. Therefore, roots would face selection pressure in optimizing host symbionts or enhancing chemical protection[45], leading to trade-offs between mycorrhizal dependency and chemical defence. Our findings, in parallel with those of Xia et al.[24], indicated that roots of AM species would maintain a low level of condensed tannins if they rely strongly on symbiotic partners for acquiring nutrients, and vice versa.

In contrast to AM species, we found a synergy between the reliance of EcM species on mycorrhiza and root chemical defence, indicated by the positive relationships of the dominant traits (BI, Supplementary Fig. 5) and the BI-RD axis scores (the PC2 scores were made negative) with the concentration of condensed tannins (Fig. 3f). This coordination could be fundamentally driven by the evolutionary shift in the root-fungal symbiosis interface from AM to EcM associations[16]. Increasing control of associations by the host reflects an important aspect of mycorrhizal evolutionary trend[16]. Unlike AM species, EcM angiosperms increase the complexity of the symbiotic interface by developing extracellular symbiotic pathways and enhance colonization sites by greater root branching[18,28,46]. To constrain the possible cheating behaviour from mycorrhizal fungi (i.e., low nutrient delivery and/or high photosynthate demand)[16], EcM hosts can modify the exodermis cells by forming suberized Casparian bands[46] and depositing phenolic compounds such as condensed tannins to restrict the further penetration of EcM fungi into the inner cortex[46–48]. These specialized features could be associated with local cell death (in and around the initial infection site) and the release of toxins and phenols compounds (e.g., tannins) caused by 'hypersensitive response' during the formation of root-fungal symbiosis[30,47–49]. Overall, this finding implies that as roots collaborate closely with mycorrhizal fungi, they will also exhibit high levels of chemical defence in EcM angiosperms. Nonetheless, further investigation is needed to examine how symbiosis and defence interact to affect the root cell wall at the microscale.

Aside from mediating mycorrhizal symbiosis, chemical defence components are also important determinants of root decomposition[27,50]. Consistent with the results from a previous study[27], condensed tannins had a negative relationship with root decomposition, irrespective of the mycorrhizal type (Figs. 2c and 3c, g), indicating their important role in slowing root decomposition. Furthermore, root-derived condensed tannins, particularly in EcM species, can form stable complexes with compounds from fungal polymers (e.g., chitin and melanin)[33]. Hence, the encased root-derived organic matter may have little opportunity to interact with extracellular enzymes[34], leading to slow root decomposition. In addition to condensed tannins, lignin, bound phenolics, and other non-structural secondary compounds are also important chemical traits driving root decomposition[24,27]. Further incorporating these chemical drivers would thus help give a more comprehensive picture of the afterlife effects of root defence within the ADD framework. Notably, dominant root traits on the symbiosis gradient, particularly RD, also largely influenced root decomposition. We found a consistently positive relationship between RD and root decomposition rate for both AM species ($R^2 = 0.097$, $P = 0.012$, Supplementary Fig. 4) and EcM species ($R^2 = 0.262$, $P = 0.009$, Supplementary Fig. 5). Interestingly, a significant negative correlation occurred between the root diameter and condensed tannin concentration, which was robust across mycorrhizal types (Supplementary Figs. 3–5). Such a tight coupling among RD, condensed tannins, and root decomposition further supported the ADD conceptual framework. Although evidence is growing that RD along with BI and CT are closely associated with mycorrhizal symbiosis[24,39,51], it is still necessary to further examine how a suite of traits with more direct links with the intensity of symbiotic associations (e.g., colonization rates and mycorrhizal C investments) fit within the ADD framework, particularly over long-time scales.

The positive correlations between root 'decomposition' and 'acquisition' depicted here agree with plant–soil feedback theories[38,52] and the MANE hypothesis[9]. The strong linkage between root decomposition rates and the RD/CT-BI axis scores, as opposed to a lack of relationship between root decomposition and the SRL-RTD axis scores (Fig. 3, Supplementary Table 6), suggests a major role for plant mycorrhizal association type in soil nutrient cycling, beyond typical indicators of root physical defences such as lignin and dry matter content[8]. In turn, the positive linkage between root decomposition and

acquisition relying more on AM fungi may induce a positive feedback loop that favours the presence of AM tree species. This strong reliance on AM fungi could be related to the lower cost and higher benefit of producing AM hyphae relative to constructing roots[25]. Compared to roots, hyphae grow faster and hence benefit more from quicker access to ephemeral inorganic nutrient patches[9,53]. In addition, the rapid turnover of hyphae allows plants more rapid cessation of resource allocation to hyphae when nutrient hotspots become depleted[54,55]. Furthermore, the variation of root decomposition rate along the 'symbiosis gradient' within the AM tree species suggests a transition from a fast-cycling belowground system that relies on AM symbiosis (ancient taxa) to a slower-cycling belowground system with lower dependence on AM symbiosis (young taxa). This pattern suggests an evolutionary strategy whereby plants tend to invest more in the root itself, resulting in higher chemical defence and slower nutrient cycling.

For EcM species, no linkage was observed between the RD-BI axis scores (assumed to represent a stronger reliance of species on EcM association with increasing BI) and root decomposition rates (Fig. 3h), but there was a strong positive relationship between BI and condensed tannins and a negative relationship between condensed tannins and mass loss (Supplementary Fig. 5). According to the MANE hypothesis, EcM fungi are characterized by an organic-nutrient economy, a trait partly retained from ancestral saprophytic fungi, and thus can obtain organic-bound nutrient either by symbiosis with plants or by decomposing or stimulating the decomposition of soil organic matter[9,56]. Our results further indicate large variation among EcM tree species, with a gradient from a slow-cycling belowground system that strongly relies on EcM symbiosis to a faster-cycling belowground system with lower dependence on EcM symbiosis. Overall, considering the large variations in root traits that occur within both the AM and EcM species groups and largely overlap with the classical AM versus EcM differences (Figs. 2 and 3) and their consequences for decomposition processes, our findings imply that the nutrient-cycling mode of plant species is more likely a progressive gradient rather than the clearcut AM versus EcM association described by the MANE hypothesis.

In conclusion, the ADD conceptual framework depicts how plants coordinate root resource acquisition, chemical defence, and nutrient reutilization (nutrient release via root decomposition), thereby forming a nutrient feedback loop in the root-soil system. The general evolutionary trend in the root system towards less dependence on symbiotic AM fungi and the emergence of the EcM association has profound consequences for the multidimensional belowground strategy of resource acquisition and plant defence and for belowground nutrient cycling. We provide here some elements that further suggest a continuum between contrasting feedback loops, ranging from a loop favouring AM species (i.e., "high AM symbiosis-low defence-fast decomposition-inorganic nutrition") to a loop favouring EcM species (i.e., "high EcM symbiosis-high defence-slow decomposition-organic nutrition"). Nonetheless, recent works depicting root trait effects on soil organic matter beyond the simple metrics of decomposability suggest that both fast decomposing roots[57] and slow decomposing roots[58] can contribute to the stabilization of soil organic matter, which limits nutrient accessibility to roots and fungi and may partly decouple root litter decomposition rate and N release. These uncertainties emphasize the necessity to provide further experimental support for the development of the ADD conceptual framework, particularly based on plant-soil feedback experiments and a broader set of taxa.

## Methods

This research complies with all relevant ethical regulations. All experimentations and soil and root sampling have been approved by the local authorities. Disturbance was limited by backfilling the topsoil after sampling under the supervision of the local forest managers.

## Study area and species

The study was conducted in temperate (Sites 1 and 2) and subtropical (Sites 3 and 4) forests in China. These four sites were located at (1) Mengluan Forestry Station (41°44′N, 117°05′E) in Hebei Province; (2) Taibai Mountain (33°49′–34°10′N, 107°19′–107°58′E) in Shanxi Province; (3) Qianyanzhou Ecological Station (26°44′N, 115°03′E) in Jiangxi Province; and (4) Yangming Mountain (25°39′N, 114°18′E) in Jiangxi Province. The elevation of these sites ranges from 102 to 2100 m, with a mean annual temperature of 2.0–17.9 °C and a mean annual precipitation of 470–1587 mm[18,59]. Tree species selected at each site were the dominant species and the most important in terms of the composition of the local plant communities, with a total of 90 tree species belonging to 55 genera and 29 families. These tree species were further classified into 65 AM plants and 25 EcM plants according to the symbiotic guild assignments of Soudzilovskaia et al. [60]. Each site has a unique set of species, as described in Supplementary Data 1.

## Field sampling and processing

We conducted field sampling in the autumn of 2018 and 2020. For each species, three well-developed trees with a similar diameter at breast height were randomly selected, with a spacing of at least 10 m apart to ensure independence; thus, 270 individuals of 90 tree species were sampled. The surface mineral soil (0–20 cm) in the sampling area was loosened carefully using a pickaxe and spade, and then the intact fine roots (including at least the first five root orders) were excavated by exposing the main lateral roots, traced back to individual trees and cleaned following the approach of Guo et al. [61]. Each root sample was divided into three sets of subsamples. One set was gently washed in deionized water and then immediately fixed in formalin-aceto-alcohol (FAA) solution (90 ml of 50% ethanol, 5 ml of 100% glacial acetic acid, and 5 ml of 37% methanol) for later anatomical measurements[18,62]. The other two sets were sealed into valve bags and then stored at −20 °C for later laboratory analysis (morphological and chemical analyses) and decomposition experiments. During root sampling, samples of surface mineral soils under each tree were also collected (0–20 cm; approx. 1 kg) and sieved (2 mm mesh size). Soil samples were sealed in valve bags, and then stored at −4 °C for later root decomposition experiments. Soil samples were evenly mixed for each site to represent the standard soil of the local site for the decomposition experiment.

## Root trait measurement

The root samples were dissected into branching orders using fine forceps following the procedure described by Pregitzer et al. [63]. The 1st- and 2nd-order roots were selected as absorptive roots because they are primarily involved in the uptake of soil resources[64].

The first set of root subsamples (1st- and 2nd-order roots) was scanned using an Epson Expression 10000 XL scanner (Seiko Epson Corporation, Suwa, Nagano, Japan) at a resolution of 400 dpi. Root diameter, length, and volume were measured using WinRHIZO Arabidopsis version 2012b (Regents Instruments Inc., Quebec, Canada). Root volume was recalculated from WINRHIZO outputs as the sum of averages of all diameter classes[50]. Afterwards, the scanned samples were oven-dried (60 °C, 48 h) and weighed to calculate SRL (root length/dry root mass) and RTD (dry root mass/root volume). The branching intensity of absorptive roots was calculated as the total number of their root tips divided by the total root length of the 3rd-order roots[18]. The oven-dried root samples were ground to a fine powder using a Retsch MM 400 mixer mill (Retsch GmbH, Haan, Germany) to determine their chemical properties. Root N concentration was measured using a Vario Macro cube elemental analyser (Elementar Analysensysteme GmbH, Langenselbold, Germany). Root condensed tannins were measured according to the acid-butanol assay[65]. First, 50 mg of sample was extracted with methanol for 30 min in a sonicator and centrifuged, and the supernatant was then analysed colourimetrically to determine the concentrations of condensed

tannins (i.e., the extractable condensed tannins). Second, the insoluble residues were freeze-dried, resuspended in methanol, incubated at 95 °C for 40 min, and then determined colourimetrically for 'nonextractable condensed tannin'. Total condensed tannins representing the sum of the extractable and nonextractable fractions, were used in subsequent analyses. Details on the extraction and analysis procedures are available in Coq et al. [65]. Proanthocyanidins were used as standards in this study. Since no common optimal standard is used for the quantification of condensed tannins and different condensed tannin structures can react differently to the assay[66], the quantification performed in this study should be interpreted as a relative estimation of the condensed tannin concentration rather than an absolute quantification. In addition, Coq et al. [65] found that the condensed tannins determined by the acid butanol method correlated reasonably well with those in leaf litter ($r = 0.93$) determined by high-performance liquid chromatography.

The second set of root subsamples (1st- and 2nd-order roots) was used to determine anatomical traits. Specifically, more than 20 root segments were randomly selected, stained with safranin-fast green, immersed in a sequence of alcohol solutions for dehydration, and then embedded in paraffin[61]. Root slices with a thickness of 8 µm were prepared and photographed with a compound microscope (DM2500, DFC450, Leica, Weztlar, Germany). Then, cortex thickness was measured using ImageJ (NIH Image, Bethesda, MD, USA).

## Root decomposition experiment

To determine the short-term decomposability of absorptive roots (1st- and 2nd-order roots), we employed a laboratory decomposition experiment in microcosms under standardized conditions[59]. Root substrates were weighed (0.200 ± 0.003 g, dry weight) after oven drying (60 °C, 48 h) and sealed in nylon litterbags (7 × 7 cm, 0.1-mm mesh). Each microcosm (plastic jar made of polyethylene terephthalate, 6.5 cm height × 8.5 cm diameter) was filled with soil (100.000 ± 0.003 g). Each litter bag was then randomly embedded in the corresponding microcosm (1 cm beneath the surface), which was filled with homogenized fresh soil from their collected site (i.e., roots from Site 1 decomposed only in the homogenized soil of Site 1, with no reciprocal transplant). Site differences in mean root decomposability were accounted for statistically, as described below. Thus, we set up a total of 270 microcosms (90 tree species × 3 replicates). All microcosms were incubated in an LHS-25SC incubator (Yiheng, Shanghai, China) in the dark at 25 ± 0.02 °C and watered with deionized water every 2 weeks to maintain soil humidity at 60% water-holding capacity during the whole experiment[59,67]. The lid of each plastic jar was drilled with a uniform number of small holes to maintain air humidity and allow for gas exchange.

All microcosms were harvested 12 weeks after the beginning of the experiment[59,68]. The mass loss of root litters ranged from 6.8% to 35.9% with a mean value of 16.7% (Supplementary Table 1), which represents the early phase of root decomposition in soil and corresponds to at least one year decomposition in the field[27,58,69]. The litterbags were opened gently and carefully after the adherent soil particles and other extraneous material were removed using a brush and forceps. The retrieved root samples were oven-dried (60 °C, 48 h) to a constant weight to determine the residual mass.

The percentage mass loss (ML) was calculated as follows:

$$ML\,(\%) = (M_b - M_a)/M_b \times 100 \qquad (1)$$

where $M_b$ and $M_a$ are the dry weight of the root litter before and after incubation, respectively.

## Data analyses

All statistical analyses were carried out using R software[70] (v.4.1.3; https://www.R-project.org/). For each tree species, we first calculated

the species mean trait values (derived from three individuals). These trait data were then $\log_{10}$-transformed to satisfy the assumption of normal distribution before analysis. The phylogenetic tree (Supplementary Fig. 1) was constructed using the 'V. PhyloMaker' package in R[71] for all tree species and AM and EcM plants separately. The 'V. PhyloMaker' package designed to generate vascular plant phylogeny, which uses the MEGA trees that are mainly derived from GBOTB (Gene Bank taxa with backbone provided by Open Tree of Life version 9.1) for seed plants[72]. Following the analysis of Wikström et al[73]. based on molecular clock theory using calibrated gene sequences and ACCTRAN optimizations, we defined the geological time when a family and its nearest sister family diverged from their immediate ancestor as the divergence time for this family. The phylogenetic signals of root traits were assessed using Blomberg's $K$ value via the 'phytools' package in R[74,75]. A larger $K$ value indicates greater phylogenetic conservatism for a trait. For all tree species, AM and EcM tree species, the variation in each root trait was estimated by calculating the means, minima and maxima and the coefficient of variation, with some root traits showing phylogenetic conservation (Supplementary Tables 1–2).

To reduce the potential influences of abiotic conditions at each site and assure variance homogeneity, we Z transformed each trait to a mean of 0 and an SD (standard deviation) of 1, for each site separately. We then used Z-scores of these traits in subsequent analyses. Bivariate relationships among root traits were assessed using the OLS regression via the 'stats' package and PGLS regression via the 'caper' package[76,77]. Considering the influence of phylogeny on root traits, pPCA was performed on all 90 tree species by using the 'phyl. pca' function in the 'phytools' package in R[78], to determine the main axes of variation among root economic traits. The pPCA was performed for six key root traits, RD, SRL, RTD, BI, CT, and root N concentration because these traits involve strategic trade-offs of resource acquisition and root-fungal symbiosis[18,26]. The pPCA was also performed by adding another two root traits (i.e., condensed tannins and decomposition rate) to test their covariation with dominant axes. One-way ANOVA was used to test the effect of mycorrhizal associations (AM vs. EcM) on the 'symbiosis gradient' (PC1) and the 'lifespan gradient' (PC2) based on pPCA results for all tree species (Supplementary Table 4). Further pPCA was conducted for AM ($n = 65$) and EcM ($n = 25$) tree species. The OLS regression was used to test relationships of the loading scores on the 'lifespan gradient' (PC1) and 'symbiosis gradient' (PC2) with root condensed tannins and root decomposition, performed for all tree species and for AM and EcM species. The OLS model was also used to test relationships of the divergence time with the loading scores of the first two axes of pPCA and root traits for major taxonomic groups at the family level (only families with at least three species were retained).

### Reporting summary

Further information on research design is available in the Nature Portfolio Reporting Summary linked to this article.

## Data availability
The raw and processed root trait data generated in this study can be accessed at https://doi.org/10.5061/dryad.p2ngf1vzn. Source data are provided with this paper.

## Code availability
The R scripts used for the analysis in this study are openly available at https://doi.org/10.5061/dryad.p2ngf1vzn.

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

## Acknowledgements

This study was financially supported by the National Natural Science Foundation of China (32222059 and 31988102, to L.K.; 32330071, to H.W.; 41830646 to S.L.), and by the 'Laboratoires d'Excellence' (LabEx) TULIP (ANR-10-LABX–41, to G.T.F.). We are grateful to Richard P. Phillips for helpful comments and suggestions.

## Author contributions

L.K. and J.Z. conceived the ideas and designed the study. J.Z. collected and analyzed the data and drew the graphics. G.T.F. and L.T. offered thoughts on the analysis and graphics. L.K., J.Z., and G.T.F. led the writing of the manuscript. L.T., S.L., H.Y., L.J., H.W., N.M., X.D. and X.F. contributed substantially to the revisions of the draft.

## Competing interests

The authors declare no competing interests.

## Additional information

¹Key Laboratory of Ecosystem Network Observation and Modeling, Institute of Geographic Sciences and Natural Resources Research, Chinese Academy of Sciences, Beijing 100101, China. ²Qianyanzhou Ecological Research Station, Key Laboratory of Ecosystem Network Observation and Modeling, Institute of Geographic Sciences and Natural Resources Research, Chinese Academy of Sciences, Beijing 100101, China. ³College of Resources and Environment, University of Chinese Academy of Sciences, Beijing 100049, China. ⁴Theoretical and Experimental Ecology Station, CNRS, 09200 Moulis, France. ⁵Mycology and Microbiology Center, University of Tartu, Tartu, Estonia. ⁶Department of Zoology, College of Science, King Saud University, Riyadh, Saudi Arabia. ⁷Freie Universität Berlin, Institut für Biologie, 14195 Berlin, Germany. ⁸Berlin-Brandenburg Institute of Advanced Biodiversity Research, 14195 Berlin, Germany. ⁹State Key Laboratory of Subtropical Silviculture, Zhejiang A&F University, Hangzhou, China. ✉e-mail: koul@igsnrr.ac.cn

