## [Peer Review File · Nature Communications]

A trait-based root acquisition-defence-decomposition framework in angiosperm tree speciesREVIEWER COMMENTS

Reviewer #1 (Remarks to the Author):

Review NCOMMS-23-56952-T

A trait-based root acquisition-defence-decomposition framework in angiosperm tree species

I would like to start this review by congratulating the authors on an absolutely brilliant manuscript. I have been looking forward exactly this level of advancement of root trait theory since the Bergmann et al. 2020 SciAdv paper. ‘You had me at’ “Since the early Devonian” in the introduction! Every time I found a point I had issue with, I went back through the manuscript and figures finding support for the authors’ point as stated. I thoroughly enjoyed the text and the figures are excellent as well—Fig. 1a is amazing and Fig 2 brings everything together very nicely!!

- 1. Key results:** The authors introduce a significant advancement of the Root Economics Spectrum (Bergmann et al. 2020) by adding additional dimensions and the Mycorrhizal Associated Nutrient Economy hypothesis (Phillips et al. 2013) by incorporating carbon and nutrient cycling feedback loops with the root trait framework. The Acquisition-Defence-Decomposition (ADD) framework proposes an interlinked root strategy whereby trade-offs in the root traits related to nutrient uptake, uses of these nutrients in chemical defences to maintain long lifespan, and the consequences on decomposition rates and subsequent release of nutrients for uptake by the plant create a feedback loop of carbon and nutrient cycling. Furthermore, this framework is highly linked to evolutionary trajectories of root development from early arbuscular mycorrhizal (AM) magnoliids to more recently diverged rosids, particularly the ectomycorrhizal (EM) angiosperms.
- 2.** There are no major flaws to the manuscript.
- 3. Originality and significance:** This manuscript represents a critical stepping stone in the evolution of our understanding for root ecology and its contribution to ecosystem ecology, by bringing together two well-established hypotheses in each of the representative fields: Root Economic Spectrum and the Mycorrhizal Associated Nutrient Economy. This theoretical advancement is supported by a robust dataset, establishing a new integrative belowground framework, which will be of interest to a range of ecologists and earth system scientists.
- 4. Data & methodology:** The data represents a major undertaking in both field and lab experiments and is will summarised in figures and the supplementary tables. Some suggestions to further improve the data and methods:

L408-410. Please add a supplementary table listing species, mean +/- SE, and which site(s) the trees were collected from (see comment on statistics). Also, this dataset is tremendous. Please consider contributing the raw or species means data to the Fine Root Ecology Database (FRED).

L430-432. I support the use of a “site-specific” soil medium for the decomposition experiment. I’m not sure if the diehard “plant-soil feedback loop” folks would approve or would have preferred some elaborate and overly complicated experiment. However, the methods used in the experiment and throughout are an appropriate compromise for the difficulty in excavating roots for such a high number of species AND the number of analyses/traits measured.

L468-474. Start with this set for consistency with the previous paragraph. What root orders were the anatomical traits measured on? Be clear on root orders used for each of the analyses.

5. Appropriate use of statistics and treatment of uncertainties (if applicable):

Overall, the statistics and reporting of uncertainties seems appropriate. However, more details could improve the appropriateness of the stats. For example, should the one-way ANOVAs be linear mixed models to control for site effects? How were the species sampled across the four sites? See my comment above about adding this information in a supplementary table.

One of the improvements over the Bergmann et al. 2021 is the use of more than 4 variables (particularly the use of CT) in the PCAs. It also resulted in a slightly different set of axes (which is not really discussed in the manuscript, but is interesting). However, what happens if you throw all the variables in a PCA? Do the variables map on the proposed ADD “axes”? This might be something to throw in a supplementary figure.

- 6. Conclusions:** Conclusions and interpretations are supported by the data and corroborated by other studies. However, because the data is represented in several ways (ie., mycorrhizal type, family, species, etc.), it does become a bit confusing to link some of the concluding statements with the results. For example, some of the phylogenetic analyses differ from the trends with divergence time, which is difficult to wrap you head around sometimes. Also, some of the overall trends differ from the mycorrhizal type specific results, so a couple words to distinguish these could be helpful (in abstract and discussion).
- 7. Suggested improvements:** This is an outstanding dataset, coupling root traits, chemical analyses, and decomposition experiments. If it is available, adding any mycorrhizal data (ie., colonization rates) would take this manuscript to the next level.
- 8. References:** The origin of plant roots is a widely studied topic, with more recent papers giving insight to the early evolution of plant roots, so I was a bit surprised by the first citation. Some alternatives (or additions) might include:

Raven JA, Edwards D. 2001. Roots: evolutionary origins and biogeochemical significance. *Journal of Experimental Botany* **52**: 381–401.

Kenrick P, Strullu-Derrien C. 2014. The Origin and Early Evolution of Roots. *Plant Physiology* **166**: 570–580.

Hetherington AJ, Dolan L. 2018. Stepwise and independent origins of roots among land plants. *Nature* **561**: 235–238.

- 9. Clarity and context:** The manuscript is exceptionally well-written with clear main points and conclusions. The explanation of the “ADD” framework in the introduction is very clear.
- 10. As an ecologist focusing on belowground research and the intersection of root ecology and biogeochemistry, everything was very clear unless otherwise noted. I think the plain language and accessible explanations will also make this manuscript clear for non-experts as well.**

Additional suggestions to further improve the manuscript

L158-1632. The hypothesis as currently stated is essentially listing the same three causal agents in both clauses. However, the hypothesis very eloquently stated in the abstract (L43-46) “we suggest a continuous shift in feedback loops, from “high AM symbiosis-low defence-fast decomposition-inorganic nutrition” by evolutionarily ancient AM species to “high EcM symbiosis-high defence-slow decomposition-organic nutrition” by more-modern EcM species”. Consider rephrasing this statement into a gradient hypothesis for a strong finish to an otherwise outstanding introduction.

L332-L335. This is a hugely underrated sentence. The observation of the direct linkages between RD, CTs, and decomposition is indeed a tighter explanation for the ADD framework than invoking mycorrhizal relationships (without any data on mycorrhizal investment or fungal traits). Rather than follow this strong point by negating it, why not strengthen it by encouraging the root trait community to start collecting data on the mycorrhizal symbiosis itself (ie., colonisation rates at a minimum, carbon investment at best, along with incorporating a suite of fungal traits relating to the ADD axes). At the end of the day there is really only one paper that strongly shows the relationship between RD/CT and mycorrhizal colonization (Xia et al. 2021) and this is for AM species, despite the explosion of papers relying on the assumption that RD is a proxy for mycorrhizal symbiosis.

L371-376. Another excellent summary sentence!! I do wonder if these underrated sentences should be highlighted more by starting the paragraphs off by these simple statements that really underline the key advancements of the ADD compared to the dichotomies of RES and MANE. Again, without more data on the fungal symbionts, do we really know if the patterns presented represent a strong to low dependence on EM or simply associations with a different EM fungal guilds?

Extended data Fig 1. Please use full genus and species epithet, especially since this information is not listed elsewhere.

Thank you again for submitting such an excellent manuscript for me to review. Hope this moves through quickly and we see it in press in the new year.

Cheers,

Dr. Kelly M. Andersen

Reviewer #2 (Remarks to the Author):

This is an interesting study that explored the neglected links among root acquisition, defense and decomposition. It is conspicuous to observe two contrasting such links between AM and EM roots which can be implied from previous studies while not verified in empirical evidence. The ms is well-written and the story is really intriguing for root ecologists and biologists working on plant-microbe interactions. While I have found key issues regarding the methods of this study which may preclude extrapolation of the results of this study to the global pattern. Importantly, there are only 90 woody species with a very limited range of absorptive root diameter, which can further constrain the implication for a global view of the root acquisition-defense-decomposition linkage. I can express my points details below:

First, the authors used live roots which is ok for the acquisition and defense study while is invalid for studying root decomposition. In leaves, there is a significant resorption of nutrients along with root senescence. Therefore, we should not use the green leaves for decomposition experiment as there is a great difference in nutrients and other chemical due to the universal resorption process in terrestrial plants. Similarly, there is nutrient resorption in roots. Therefore, the authors should not use live roots for decomposition study. Although the authors could argue that decomposition studies using live roots is popular in root ecological studies, this method is wrong for the topic of the current study. This is because the authors aimed to explore the linkage of processes between live roots (acquisition and defense) and dead roots (decomposition). While the authors used live roots for decomposition, a case not fulfilling the purpose of this study.

Second, the authors used standardized lab microcosms to examine the root decomposition. This is sharply different from the conditions in the natural conditions with abundant soil microbes and soil animals for decomposing the roots. More importantly, the authors studied root decomposition in subtropical and temperate forests with very different soil and climatic conditions, while the authors as their method section goes use the similar lab conditions. This empirical schedule great limit the applications of this study to the field condition. In other words, we don't know whether the acquisition-defense-decomposition linkage is the same as it is in the field.

Third, the decomposition experiment only lasted 12 weeks (about 3 month), a very short term compared with other studies on root decomposition usually lasting 2 to more years. To answer for whether there any loop or feedback in the ADD framework, it is necessary to test

this framework using the longer root decomposition study and a field condition.

Fourth, it is interesting to note an almost decoupling relationship between RD and SRL, which has been reported to be strong and negatively correlated in a wide range of study from many case studies to global studies. This unexpected RD-SRL relationship is due to a very limited range of absorptive root diameter, from 0.32 to 0.60 mm. While there is much greater variation of the root diameter, e.g., almost 100-fold variation of the absorptive root diameter from the data in the classical FRED dataset. Therefore, the patterns regarding RD-SRL and root decomposition relationships are not convinced. The authors should test the ADD framework by sampling plant species spanning a large rather than a very small range of absorptive root diameter.

Fifth, the authors focus mainly on the role of tannin in root decomposition, while neglecting the other key chemicals that strong affecting root decomposition, such as lignin, non-structural carbon content, N content. We are not sure whether there is covariation of the tannin with lignin and other chemicals. If so, the authors should exclude the co-influence of these non-tannin chemicals on root decomposition to verify their ADD hypothesis.

Sixth, since the authors have examined root anatomy, they can incorporate the data on mycorrhizal colonization in this study as the symbiont dimension, the key trait dimension in this study, describe the mycorrhizal association. As for EM roots, I suggest the authors could give the exact picture or date on how the ECM fungi affect on the cell walls of the roots. This is a direct evidence for the authors' argument of their hypothesis regarding the sharp difference between AM and EM roots in their root-mycorrhizae interaction.

Response to the reviewers' comments:

'A trait-based root acquisition-defence-decomposition framework in angiosperm tree species'

General remarks

We greatly appreciate the two reviewers for offering us constructive and insightful comments, which were extremely helpful for us to improve our manuscript.

We have revised the manuscript following various valuable comments provided (highlighted in yellow) and believe that this resulted in a more accurate, detailed and balanced account of our work.

Below we provide point by point responses to the reviewer's comments. Please note that the comments from the reviewers are in regular text followed by our responses in blue text and the revisions in "*blue and italic*" text, when necessary.

Point-by-point responses to reviewers' comments

Reviewer #1 (Remarks to the Author):

Dear Authors,

I would like to start this review by congratulating the authors on an absolutely brilliant manuscript. I have been looking forward exactly this level of advancement of root trait theory since the Bergmann et al. 2020 SciAdv paper. 'You had me at' "Since the early Devonian" in the introduction! Every time I found a point I had issue with, I went back through the manuscript and figures finding support for the authors' point as stated. I thoroughly enjoyed the text and the figures are excellent as well—Fig. 1a is amazing and Fig 2 brings everything together very nicely!!

1. Key results: The authors introduce a significant advancement of the Root Economics Spectrum (Bergmann et al. 2020) by adding additional dimensions and the Mycorrhizal Associated Nutrient Economy hypothesis (Phillips et al. 2013) by incorporating carbon and nutrient cycling feedback loops with the root trait framework. The Acquisition-Defence-Decomposition (ADD) framework proposes an interlinked root strategy whereby trade-offs in the root traits related to nutrient uptake, uses of these nutrients in chemical defences to maintain long lifespan, and the consequences on decomposition rates and subsequent release of nutrients for uptake by the plant create a feedback loop of carbon and nutrient cycling. Furthermore, this framework is highly linked to evolutionary trajectories of root development from early arbuscular

mycorrhizal (AM) magnoliids to more recently diverged rosids, particularly the ectomycorrhizal (EM) angiosperms.

2. There are no major flaws to the manuscript.

3. Originality and significance: This manuscript represents a critical stepping stone in the evolution of our understanding for root ecology and its contribution to ecosystem ecology, by bringing together two well-established hypotheses in each of the representative fields: Root Economic Spectrum and the Mycorrhizal Associated Nutrient Economy. This theoretical advancement is supported by a robust dataset, establishing a new integrative belowground framework, which will be of interest to a range of ecologists and earth system scientists.

Response to Q1-3: We are grateful to your favourable remarks and thank you very much for highlighting the strengths and the novelty of our study.

4. Data & methodology: The data represents a major undertaking in both field and lab experiments and is will summarised in figures and the supplementary tables. Some suggestions to further improve the data and methods:

L408-410. Please add a supplementary table listing species, mean +/- SE, and which site(s) the trees were collected from (see comment on statistics). Also, this dataset is tremendous. Please consider contributing the raw or species means data to the Fine Root Ecology Database (FRED).

Response: Thank you for recognizing our efforts in both field and lab experiments, and providing helpful advice. We have added **Supplementary Table 7** including the species and site information in the Supplementary information, and we are willing to contribute the data to the FRED after publication.

L430-432. I support the use of a “site-specific” soil medium for the decomposition experiment. I’m not sure if the diehard “plant-soil feedback loop” folks would approve or would have preferred some elaborate and overly complicated experiment. However, the methods used in the experiment and throughout are an appropriate compromise for the difficulty in excavating roots for such a high number of species AND the number of analyses/traits measured.

Response: Thanks for your kind comments. We agree with your point that field sampling, particularly collecting sufficient distal absorptive roots (i.e. 1st- and 2nd-order roots), was a very challenging work, so we made an appropriate compromise. We agree that it deserves further study, particularly field experiments, to better understand the context-dependency of the ADD hypothesis proposed here.

L468-474. Start with this set for consistency with the previous paragraph. What root orders were the anatomical traits measured on? Be clear on root orders used for each of the analyses.

Response: Thanks for your thoughtful comments. We have detailed this sentence to make it clear. (Lines 507-508). *“The second set of root subsamples (1st- and 2nd-order roots) was used to determine anatomical traits.”*

Additionally, we have also made minor changes in the previous paragraph for clarity.

Lines 477-478: *“The first set of root subsamples (1st- and 2nd-order roots) was scanned using an Epson Expression...”*

Lines 516-517: *To determine the short-term decomposability of absorptive roots (1st- and 2nd-order roots)...*”

5. Appropriate use of statistics and treatment of uncertainties (if applicable): Overall, the statistics and reporting of uncertainties seems appropriate. However, more details could improve the appropriateness of the stats. For example, should the one-way ANOVAs be linear mixed models to control for site effects? How were the species sampled across the four sites? See my comment above about adding this information in a supplementary table. One of the improvements over the Bergmann et al. 2021 is the use of more than 4 variables (particularly the use of CT) in the PCAs. It also resulted in a slightly different set of axes (which is not really discussed in the manuscript, but is interesting). However, what happens if you throw all the variables in a PCA? Do the variables map on the proposed ADD “axes”? This might be something to throw in a supplementary figure.

5.1 Overall, the statistics and reporting of uncertainties seems appropriate. However, more details could improve the appropriateness of the stats.

Response: Thanks for your comments and we apologize for the confusion. To reduce the potential influences of abiotic conditions at each site, we have Z-transformed each trait to a mean of 0 and an SD (standard deviation) of 1, for each site separately, before performing further analysis. That is to say, we used Z-scores of these traits in further analyses. For example, when using ANOVA to test the effect of ‘site’ and ‘mycorrhizal type’ on the score of ‘symbiosis gradient’(PC1) and ‘lifespan gradient’(PC2) of 90 tree species, we found that there was no site effect (**Table R1**), so only the effect of mycorrhizal type on axis scores was placed in **Supplementary Table 4**. Therefore, 'Site effects' were not considered separately in the relevant analyses. We have made minor changes in the section of *Data analyses* (Lines 564-567), as well as additional notes in **Supplementary Table 2** to more clearly explain how we dealt with site effects:

Lines 564-567: *“To reduce the potential influences of abiotic conditions at each site and assure variance homogeneity, we Z transformed each trait to a mean of 0 and an SD (standard deviation)*

of 1, for each site separately. We then used Z-scores of these traits in subsequent analyses”.

We added this sentence in the notes of the **Supplementary Table 2**:

“Significant differences between means were compared using ANOVA based on Z-scores of the traits ($P < 0.05$)”.

Table R1 | ANOVA used to test the effect of site and mycorrhizal type on the score of ‘symbiosis gradient’(PC1) and ‘lifespan gradient’(PC2) of 90 tree species.

	Term	F	df	P
PC1	Site	0.000	3, 82	1.000
	Mycorrhizal	19.417	1, 82	< 0.001
	Site × Mycorrhizal	0.754	3, 82	0.523
PC2	Site	0.000	3, 82	1.000
	Mycorrhizal	3.845	1, 82	0.053
	Site × Mycorrhizal	1.961	3, 82	0.126

5.2 How were the species sampled across the four sites? See my comment above about adding this information in a supplementary table.

Response: We conducted field sampling in two temperate (Sites 1 and 2) and two subtropical (Sites 3 and 4) forests in China. All selected species are common within these four sampling areas and representative of local forest communities. Each species with three individuals, thus 270 individuals of 90 tree species were sampled. More detailed information can be found in the Method section. Following your suggestion, we have put the site information into the newly added **Supplementary Table 7** in the Supplementary Information.

5.3 One of the improvements over the Bergmann et al. 2020 is the use of more than 4 variables (particularly the use of CT) in the PCAs. It also resulted in a slightly different set of axes (which is not really discussed in the manuscript, but is interesting).

Response: Thanks for your insightful and valuable comments. We agree with your point that the RES pattern in this study differed from that in Bergman et al. (2020). The tree species in our dataset represent a subset of species (here we focus on angiosperms trees) from Yan et al. (2022). Although we have given some discussions in the previous study (Yan et al., 2022), we believe this point proposed by you and reviewer#2 is very important. Therefore, we have added a discussion paragraph to further elaborate on this issue (Lines 269-285).

“The RES revealed here differed from the axis representations in Bergman et al. (2020) where RD-SRL axis represents a “collaboration” gradient and RTD-N axis represents a “conservation”

gradient. Here, we argue that these two representations of the RES are overall consistent with each other, and the fundamental divergence stems from our sampling of tree species displaying little variation in RD. Based on the theoretical formula: $SRL = 4/(\pi \times RTD \times RD^2)$, any change in RD has much stronger weight on SRL than change in RTD because RD is quadratic to SRL (Zhang et al. 2024). In contrast to the global dataset of Bergmann et al. (2020), who used a broad definition of fine roots, our dataset is restricted to a narrow range of root orders (1st- and 2nd-order roots). It also focuses on a narrower group of plants ('trees' only rather than 'woody species' category), and lineage (angiosperms). This stringent selection is likely responsible for the narrow interspecific variation in RD, which led to the decoupling between RD and SRL. In contrast, we found a large variation in RTD across species leading to a strong coupling between SRL and RTD. Interestingly, our results, based on a strong representation of subtropical forests (60%, Supplementary Table 7), support previous observations that changes in SRL may be more strongly regulated by RTD compared to RD in tropical forests (Valverde-Barrantes et al 2021). ”

5.4 However, what happens if you throw all the variables in a PCA? Do the variables map on the proposed ADD “axes”? This might be something to throw in a supplementary figure.

Response: Very good point! Following your insightful suggestion, we have added the pPCA analysis performed on all root traits, and the results indicated that both condensed tannins and decomposition rate were located on the ‘symbiosis gradient’(PC1)(**Fig. R1, Table R2**). Specifically, we found that RD and CT were correlated positively with decomposition rate and negatively with condensed tannins, which further supports the ADD framework. We have included these results in the Supplementary information and made corresponding additions in the main text of the revised manuscript.

Lines 183-187:

“Furthermore, when pPCA was performed by including condensed tannins and decomposition rate, we found that these two traits were coaxial with RD, CT, and BI. Specifically, RD and CT were positively correlated with decomposition rate and negatively with condensed tannins (Supplementary Fig. 2; Supplementary Table 5).”

Lines 219-221:

“When condensed tannins and decomposition rate were added to the pPCA of AM species, they aligned very closely to the PC1 axis, with little overall changes in the pPCA representation (Supplementary Fig. 2; Supplementary Table 5).”

Lines 227-230:

“When condensed tannins and decomposition rate were added to the pPCA of EcM species, they aligned very closely to the PC1 axis, along with RD and BI. Specifically, BI was correlated negatively with decomposition rate and positively with condensed tannins (Supplementary Fig. 2;

Supplementary Table 5).”

Fig. R1 | The test of the root economic space (RES) using the eight traits (RD, SRL, RTD, RN, BI, CT, CTs, ML)(same as Supplementary Fig. 2 in Supplementary information). Root trait biplots of phylogenetic principal component analysis (pPCA) for (a) all 90 tree species; (b) 65 arbuscular mycorrhizal (AM) species (purple circle); (c) 25 ectomycorrhizal (EcM) species (green circle). The detailed information of pPCA results sees Table R2. Both CTs and ML are located on the ‘symbiosis gradient’(PC1). RD, root diameter; SRL, specific root length; RTD, root tissue density; RN, root nitrogen concentration; BI, branching intensity; CT, cortex thickness; CTs, condensed tannins concentration; ML, mass loss

Table R2 | Results of phylogenetic principal component analyses based on 8 root traits for all species, arbuscular mycorrhizal (AM) species, and ectomycorrhizal (EcM) species (same as **Supplementary Table 5** in Supplementary information). Displayed are the Eigenvalue and the proportion of variation explained by each principal component (PC) and the loadings of the root traits.

		PC1	PC2	PC3	PC4
All species n = 90	Eigenvalue	2.722	1.983	1.208	0.742
	Variance	0.340	0.248	0.151	0.093
	RD	0.873	-0.107	-0.274	0.080
	SRL	0.078	0.981	0.113	-0.071
	RTD	-0.237	-0.960	-0.012	0.042
	RN	0.344	-0.251	0.681	-0.515
	BI	-0.637	0.146	0.426	0.176
	CT	0.719	0.028	-0.372	-0.147
	ML	0.592	-0.053	0.425	0.639
AM species n = 65	Eigenvalue	2.907	2.022	1.278	0.717
	Variance	0.340	0.248	0.151	0.093
	RD	0.900	0.095	0.288	0.056
	SRL	0.030	0.986	-0.008	-0.111
	RTD	-0.213	-0.962	-0.080	0.080
	RN	0.351	-0.178	-0.676	-0.588
	BI	-0.709	0.274	-0.313	0.131
	CT	0.764	-0.013	0.423	-0.086
	ML	0.533	0.081	-0.554	0.568
EcM species n = 25	Eigenvalue	2.322	2.114	1.208	0.918
	Variance	0.290	0.264	0.151	0.115
	RD	0.722	0.214	0.107	0.092
	SRL	-0.410	0.820	-0.125	0.226
	RTD	0.308	-0.897	-0.026	-0.214
	RN	0.149	0.171	-0.744	-0.570
	BI	-0.472	-0.187	-0.686	0.259
	CT	0.001	0.661	0.218	-0.502
	ML	0.759	0.205	-0.309	0.410

RD, root diameter; SRL, specific root length; RTD, root tissue density; RN, root nitrogen concentration; BI, branching intensity; CT, cortex thickness; CTs, condensed tannins concentration; ML, mass loss.

6. Conclusions: Conclusions and interpretations are supported by the data and corroborated by other studies. However, because the data is represented in several ways (ie., mycorrhizal type, family, species, etc.), it does become a bit confusing to link some of the concluding statements with the results. For example, some of the phylogenetic analyses differ from the trends with divergence time, which is difficult to wrap you head around sometimes. Also, some of the overall trends differ from the mycorrhizal type specific results, so a couple words to distinguish these could be helpful (in abstract and discussion).

Response: Thanks again for your insightful comments. We apologize for the mismatch between the *K*-statistic and the *p*-value tested in the original version. After rechecking our data analysis, we found that the *p*-value provided in the original version was incorrect because the "multiphylosignal" function we used only reported Blombergs' *K*, PIC.variance, and the *p*-value for PIC.variance but not the *p*-value for Blombergs' *K*. In the revised manuscript, we have reanalyzed by using the "phylosig" function, which can output both Blombergs' *K* and its corresponding *p*-value (see **Supplementary Table 1**). Considering our analysis related to divergence time was based on the family level for the major taxa ($n = 15$, only families with at least three species were retained), we thus performed phylogenetic analysis for tree species ($n = 70$) from these major taxa and added these results in the revised manuscript (**Table R3**; see Lines 169-171 in main text). We found that RD, CT, concentrations of tissue N and condensed tannins showed significant phylogenetic signals, and these results overall support the relationships between root traits and divergence time. In addition, to avoid distracting the readers, we have removed the results on the phylogenetic signals after distinguishing mycorrhizal types in the revised manuscript, because these results seem not necessary in the current frame. Following your suggestion, we have also added a couple of words (e.g. species level or family level) where necessary to make it clear.

Table R3 | Blombergs' *K* value of root traits for 70 tree species from 15 families (integrated into **Supplementary Table 1** in Supplementary information).

Traits	K -value	P -value
RD	0.086	0.025
SRL	0.015	0.415
RTD	0.020	0.320
BI	0.039	0.089
CT	0.066	0.036
RN	0.081	0.007
CTs	0.100	0.004
ML	0.043	0.097

RD, root diameter; SRL, specific root length; RTD, root tissue density; BI, branching intensity; CT, cortex thickness; RN, root nitrogen concentration; CTs, condensed tannins concentration; ML, mass loss. The significant *K* values are in bold ($P < 0.05$).

7. Suggested improvements: This is an outstanding dataset, coupling root traits, chemical analyses, decomposition and experiments. If it is available, adding any mycorrhizal data (ie., colonization rates) would take this manuscript to the next level.

Response: Thanks for the comment. It was in our plan to measure mycorrhizal colonization intensity (MCI), but a substantial part of the data was accidentally lost during the campaign and the few species MCI data available are poorly representative of the full species dataset. Considering that the colonization rate is largely influenced by seasonal variation and sampling location/soil microenvironment (Brundrett, 1991; Heinemeyer et al., 2004), we did not measure this trait in the subsequent sampling. Similar to our previous study (Yan et al., 2022), we used the dominant traits (RD, CT, and BI) in conjunction with the axis scores of ‘symbiosis gradient’ as a proxy of fungal-symbiosis dependency in root strategy. We have now explicitly pointed out the necessity to further incorporate direct mycorrhizal symbiotic traits (colonization rate, carbon investment, and a series of mycorrhizal traits) within the ADD framework in the future. (Lines 374-377).

“... it is still necessary to further examine how a suite of traits with more direct links with the intensity of symbiotic associations, such as colonization rates and mycorrhizal C investment, fits within the ADD framework...”

8. References: The origin of plant roots is a widely studied topic, with more recent papers giving insight to the early evolution of plant roots, so I was a bit surprised by the first citation. Some alternatives (or additions) might include:

Raven JA, Edwards D. 2001. Roots: evolutionary origins and biogeochemical significance. *Journal of Experimental Botany* 52: 381–401.

Kenrick P, Strullu-Derrien C. 2014. The Origin and Early Evolution of Roots. *Plant Physiology* 166: 570– 580.

Hetherington AJ, Dolan L. 2018. Stepwise and independent origins of roots among land plants. *Nature* 561: 235–238.

Response: Thank you very much for providing these valuable and appropriate references, which have been cited in the revised manuscript (Line 53).

9. Clarity and context: The manuscript is exceptionally well-written with clear main points and conclusions. The explanation of the “ADD” framework in the introduction is very clear.

Response: We greatly appreciate you for the positive evaluation of our manuscript.

10. As an ecologist focusing on belowground research and the intersection of root ecology and biogeochemistry, everything was very clear unless otherwise noted. I think the plain language and accessible explanations will also make this manuscript clear for non-experts as well.

Response: We sincerely appreciate your acknowledgment of our study. We have polished the language as much as we could in the revised manuscript, to make it more concise and readable.

Additional suggestions to further improve the manuscript:

L158-1632. The hypothesis as currently stated is essentially listing the same three causal agents in both clauses. However, the hypothesis very eloquently stated in the abstract (L43-46) “we suggest a continuous shift in feedback loops, from “high AM symbiosis-low defence-fast decomposition-inorganic nutrition” by evolutionarily ancient AM species to “high EcM symbiosis-high defence-slow decomposition- organic nutrition” by more-modern EcM species”. Consider rephrasing this statement into a gradient hypothesis for a strong finish to an otherwise outstanding introduction.

Response: Thank you very much for the insightful comments. Following your suggestion, we have rephrased this hypothesis as: *“Considering the distinct evolutionary histories, symbiotic pathways, and nutrient economies of AM and EcM species^{9,25} and large variation in trait values within each mycorrhizal type, we further hypothesized that evolutionary shifts from AM to EcM association mediate the ‘acquisition-defence-decomposition’ linkages, thus forming a progressive gradient of nutrient-cycling modes.”* (Lines 159-162)

L332-L335. This is a hugely underrated sentence. The observation of the direct linkages between RD, CTs, and decomposition is indeed a tighter explanation for the ADD framework than invoking mycorrhizal relationships (without any data on mycorrhizal investment or fungal traits). Rather than follow this strong point by negating it, why not strengthen it by encouraging the root trait community to start collecting data on the mycorrhizal symbiosis itself (ie., colonisation rates at a minimum, carbon investment at best, along with incorporating a suite of fungal traits relating to the ADD axes). At the end of the day there is really only one paper that strongly shows the relationship between RD/CT and mycorrhizal colonization (Xia et al. 2021) and this is for AM species, despite the explosion of papers relying on the assumption that RD is a proxy for mycorrhizal symbiosis.

Response: Thanks for your insightful comments. We agree with your point that further research needs to incorporate more directly mycorrhizal-related traits to further elucidate the links within ADD. Therefore, we have changed the phrasing here and added some discussions about this point (Lines 372-377).

“Such a tight coupling among RD, condensed tannins, and root decomposition further supported the ADD conceptual framework. Although evidence is growing that RD along with BI and CT are

closely associated with mycorrhizal symbiosis (Xia et al., 2021; Xia 2022; Ven et al., 2019), it is still necessary to further examine how a suite of traits with more direct links with the intensity of symbiotic associations (e.g. colonization rates and mycorrhizal C investments), fits within the ADD framework...”

L371-376. Another excellent summary sentence!! I do wonder if these underrated sentences should be highlighted more by starting the paragraphs off by these simple statements that really underline the key advancements of the ADD compared to the dichotomies of RES and MANE. Again, without more data on the fungal symbionts, do we really know if the patterns presented represent a strong to low dependence on EM or simply associations with a different EM fungal guilds?

Response: We agree with your point that this paragraph is a summary sentence of this study, while considering that the previous paragraphs follow the logic of ‘A-D, D-D, D-A’, thus we thought it would be more logical and persuasive to provide an overall summary of the ‘ADD’ process after finishing the discussion about ‘D-A’. We have now highlighted this sentence more clearly by ending the paragraph with this statement (Lines 411-416).

In addition, we emphasized in the Discussion the necessity for obtaining in the future more direct data on fungal symbionts to verify the internal linkages of ADD. (Lines 374-377)

“...it is still necessary to further examine how a suite of traits with more direct links with the intensity of symbiotic associations (e.g. colonization rates and mycorrhizal C investments), fit within the ADD framework....”

Supplementary Fig 1. Please use full genus and species epithet, especially since this information is not listed elsewhere.

Response: Thanks for the helpful advice. We have updated **Supplementary Fig. 1** (in Supplementary information) as you suggested, which now includes complete information about the species.

Thank you again for submitting such an excellent manuscript for me to review. Hope this moves through quickly and we see it in press in the new year.

Cheers,

Dr. Kelly M. Andersen

Response: We thank you again for providing such valuable and important feedback. We are delighted to have had such a favorable assessment of our manuscript.

References

- Bergmann, J. et al. The fungal collaboration gradient dominates the root economics space in plants. *Science Advances* **6**, eaba3756 (2020).
- Brundrett, M. Mycorrhizas in natural ecosystems. *Advances in Ecological Research* **21**, 171-313 (1991).
- Heinemeyer, A. et al. Impact of soil warming and shading on colonization and community structure of arbuscular mycorrhizal fungi in roots of a native grassland community. *Global Change Biology* **10**, 52-64 (2004).
- Valverde-Barrantes, O. J., Authier, L., Schimann, H. & Baraloto, C. Root anatomy helps to reconcile observed root trait syndromes in tropical tree species. *American Journal of Botany* **108**, 744-755 (2021).
- Ven, A., Verlinden, M. S., Verbruggen, E. & Vicca, S. Experimental evidence that phosphorus fertilization and arbuscular mycorrhizal symbiosis can reduce the carbon cost of phosphorus uptake. *Functional Ecology* **33**, 2215-2225 (2019).
- Xia, M., Suseela, V., McCormack, M. L., Kennedy, P. G. & Tharayil, N. Common and lifestyle-specific traits of mycorrhizal root metabolome reflect ecological strategies of plant-mycorrhizal interactions. *Journal of Ecology* **111**, 601-616 (2022).
- Xia, M., Valverde-Barrantes, O. J., Suseela, V., Blackwood, C. B. & Tharayil, N. Coordination between compound-specific chemistry and morphology in plant roots aligns with ancestral mycorrhizal association in woody angiosperms. *New Phytologist* **232**, 1259-1271 (2021).
- Yan, H. et al. Mycorrhizal symbiosis pathway and edaphic fertility frame root economics space among tree species. *New Phytologist* **234**, 1639-1653 (2022).
- Zhang, Y. et al. The origin of bi-dimensionality in plant root traits. *Trends in ecology & evolution* **39**, 78-88 (2024).

Reviewer #2 (Remarks to the Author):

This is an interesting study that explored the neglected links among root acquisition, defense and decomposition. It is conspicuous to observe two contrasting such links between AM and EM roots which can be implied from previous studies while not verified in empirical evidence. The ms is well-written and the story is really intriguing for root ecologists and biologists working on plant-microbe interactions.

While I have found key issues regarding the methods of this study which may preclude extrapolation of the results of this study to the global pattern. Importantly, there are only 90 woody species with a very limited range of absorptive root diameter, which can further constrain the implication for a global view of the root acquisition-defense-decomposition linkage. I can express my points details below:

First, the authors used live roots which is ok for the acquisition and defense study while is invalid for studying root decomposition. In leaves, there is a significant resorption of nutrients along with root senescence. Therefore, we should not use the green leaves for decomposition experiment as there is a great difference in nutrients and other chemical due to the universal resorption process in terrestrial plants. Similarly, there is nutrient resorption in roots. Therefore, the authors should not use live roots for decomposition study. Although the authors could argue that decomposition studies using live roots is popular in root ecological studies, this method is wrong for the topic of the current study. This is because the authors aimed to explore the linkage of processes between live roots (acquisition and defense) and dead roots (decomposition). While the authors used live roots for decomposition, a case not fulfilling the purpose of this study.

Response: Thank you for this comment. We agree with you and acknowledge that the issue is very important, but also quite challenging at the current level of observation and technology for root study. Firstly, it is difficult to detect the status of dead roots, particularly in the soil medium. Unlike the way leaves senesce, abscise, and shed discretely, the root system gradually loses its function with age (Eissenstat and Volder, 2004) and is colonized by saprophytic fungi while still alive (Resendes et al., 2008). Therefore, the difference between living roots and decomposed roots represents a continuum, making it extremely difficult to collect dead roots that have not yet begun to decompose (Hobbie et al., 2010). Secondly, the distal roots are very fragile, so even if the dead roots can be determined, it remains difficult to collect a sufficient amount of dead distal roots (Ma et al., 2015). Based on the comprehensive consideration of the above factors, with reference to previous studies, we chose living roots as a proxy considering they best represent roots that have not yet begun to decompose (Hobbie et al., 2010). We agree with you that this may bring uncertainties, and we hope that there will be a technological breakthrough in the future that will allow us to carry out root decomposition studies on the basis of dead roots and to further verify the ADD conceptual framework.

Second, the authors used standardized lab microcosms to examine the root decomposition. This is sharply different from the conditions in the natural conditions with abundant soil microbes and soil animals for decomposing the roots. More importantly, the authors studied root decomposition in subtropical and temperate forests with very different soil and climatic conditions, while the authors as their method section goes use the similar lab conditions. This empirical schedule great limit the applications of this study to the field condition. In other words, we don't know whether the acquisition- defense-decomposition linkage is the same as it is in the field.

Response: Thank you so much for your valuable comments. We fully agree that the lab experiment could not fully mimic the field conditions with considerable variability and heterogeneity of soil and climate as you mentioned, thus the microcosm incubations here represent a potential decomposition rate (namely litter 'decomposability') that can be in situ complicated by multiple environmental factors, especially the soil microbial community. Therefore, this study attempted only to provide an ADD conceptual framework based on root traits (including 'decomposability') and performed a preliminary validation against the current dataset. We noticed that it is a great challenge to test this framework in the field, thus we made an appropriate compromise and testified its applicability under similar or the same conditions. In this study, we instead performed these standardized microcosm incubation experiments (as usually did in literature, e.g. Roumet et al., 2016; Jiang et al., 2021) using the soils taken from the respective sites of root sampling considering the overarching importance of the soil substrate biotic (microbial community composition, mesofauna, etc.) and abiotic (organic matter, pH, nutrient availability, etc.) factors for assessing litter decomposition rate. Moreover, to reduce the potential influences of abiotic conditions at each site, all root traits and decomposability measurements were z-standardized to the mean and standard deviation of that trait for the site in which it was collected before performing further analysis (see Data analyses, Lines 564-567). We agree with you that further validation is really needed in the field and across a broader range of species and environmental conditions in the future although our proposed ADD framework explicitly provides a foundation for a set of testable hypotheses. This necessity has been highlighted in the revised manuscript, please see Lines 433-435.

Third, the decomposition experiment only lasted 12 weeks (about 3 month), a very short term compared with other studies on root decomposition usually lasting 2 to more years. To answer for whether there any loop or feedback in the ADD framework, it is necessary to test this framework using the longer root decomposition study and a field condition.

Response: Thank you for the insightful comments. The ADD conceptual framework proposed here focuses more on the effect of nutrient release from decomposing roots on nutrient acquisition. This effect is relatively well captured in the early stage of decomposition when all easily leached and easily decomposed nutrient-rich compounds are released. In addition, previous studies have shown that decomposition rates in microcosms are relatively fast in the first three months and correspond approximately to 1-2 years of root decomposition rate in field conditions (Sun et al., 2018; Jiang et al., 2018; Huys et al. 2022). Referring to previous studies, we conducted a three-

month microcosm decomposition experiment (Roumet et al., 2016; Jiang et al., 2021), and also found that the decomposition rate (mean value = 16.7%) after three months basically corresponds to that from the field condition for at least one year. We have further elucidated this point in the Method section (Lines 533-537). Overall, we agree that our microcosm experiment is only a first step towards describing and testing the proposed ADD conceptual framework and we now more explicitly highlight that this framework should be further tested over long-time scales (Line 376).

(Lines 533-537)

“All microcosms were harvested 12 weeks after the beginning of the experiment with reference to previous studies (Roumet et al., 2016; Jiang et al., 2021). The mass loss of root litters ranged from 6.8% to 35.9% with a mean value of 16.7% (Supplementary Table 1), which represents the early phase of root decomposition in soil and corresponds to at least one-year decomposition in the field (Sun et al., 2018; Jiang et al., 2018; Huys et al. 2022).”

Fourth, it is interesting to note an almost decoupling relationship between RD and SRL, which has been reported to be strong and negatively correlated in a wide range of study from many case studies to global studies. This unexpected RD-SRL relationship is due to a very limited range of absorptive root diameter, from 0.32 to 0.60 mm. While there is much greater variation of the root diameter, e.g., almost 100-fold variation of the absorptive root diameter from the data in the classical FRED dataset. Therefore, the patterns regarding RD-SRL and root decomposition relationships are not convinced. The authors should test the ADD framework by sampling plant species spanning a large rather than a very small range of absorptive root diameter.

Response: Thanks for your valuable comments. We think this point proposed by you and reviewer #1 is very insightful. Therefore, we have added a paragraph in the discussion section to further elaborate on this issue (see Lines 269-285).

“The RES revealed here differed from the axis representations in Bergman et al. (2020) where RD-SRL axis represents a “collaboration” gradient and RTD-N axis represents a “conservation” gradient. Here, we argue that these two representations of the RES are overall consistent with each other, and the fundamental divergence stems from our sampling of tree species displaying little variation in RD. Based on the theoretical formula: $SRL = 4/(\pi \times RTD \times RD^2)$, any change in RD has much stronger weight on SRL than change in RTD because RD is quadratic to SRL (Zhang et al. 2024). In contrast to the global dataset of Bergmann et al. (2020), who used a broad definition of fine roots, our dataset is restricted to a narrow range of root orders (1st- and 2nd-order roots). It also focuses on a narrower group of plants (‘trees’ only rather than ‘woody species’ category), and lineage (angiosperms). This stringent selection is likely responsible for the narrow interspecific variation in RD, which led to the decoupling between RD and SRL. In contrast, we found a large variation in RTD across species leading to a strong coupling between SRL and RTD. Interestingly, our results, based on a strong representation of subtropical forests

(60%, Supplementary Table 7), support previous observations that changes in SRL may be more strongly regulated by RTD compared to RD in tropical forests (Valverde-Barrantes et al 2021). ”

Overall, we would also like to emphasize that characterizing as many as 90 species of trees for two biomes and a very large range of root traits is two to three times more work than typical for this type of study and could be considered a fair dataset for laying the foundation of a novel conceptual framework such as the ADD, which will unavoidably need further confirmation on broader set of plant taxa. Given this, we have rewritten some parts of the manuscript to better emphasize that this work represents a proof-of-concept and preliminary testing for the ADD conceptual framework (e.g. Lines 433-435).

Fifth, the authors focus mainly on the role of tannin in root decomposition, while neglecting the other key chemicals that strong affecting root decomposition, such as lignin, non-structural carbon content, N content. We are not sure whether there is covariation of the tannin with lignin and other chemicals. If so, the authors should exclude the co-influence of these non-tannin chemicals on root decomposition to verify their ADD hypothesis.

Response: Thank you for the thoughtful comments. Sampling in the field, particularly the collection of the distal absorptive roots (1st- and 2nd-order roots), was indeed very difficult. Especially, our absorptive root samples were further subdivided into three sets of sub-samples for the measurement of morphological and anatomical traits, as well as the setup of the root decomposition experiment, respectively. As a result, only a small number of samples were finally used to measure chemical traits. In addition, different categories of chemical root traits are generally determined using independent methods, which further limits our ability to determine a large range of chemical traits. It has been reported that absorptive root decomposition is largely regulated by condensed tannins rather than more conventional indicators like lignin, which is thought to be a key indicator controlling leaf decomposition (Sun et al., 2018; Xia et al., 2021). Moreover, condensed tannins are key drivers of the availability of nutrients during decomposition, which is an important focus of the ADD framework, owing to their complexing abilities regarding N-rich proteins in particular. Therefore, we have chosen the key chemical defense variable—condensed tannins, instead of lignin. In addition, we also measured the root N and C concentrations. Following your suggestion, we further used partial correlation analysis to investigate the effect of CTs on decomposition in all tree species (as well as in AM and EcM tree species) by controlling for RN and C/N. The major results are as follows:

Statistically significant negative correlations were found between CTs and ML (**Fig. R2**). The results of Partial correlation analysis suggest that ML is also significantly correlated with CTs when statistically accounting for the confounding effects of RN, C/N, or both RN and C/N (**Table R4**). By contrast, the effects of RN and C/N on ML were not significant after removing the effects of CTs. These results remain robust even after distinguishing between mycorrhizal types. Overall,

these results support the ADD hypothesis that condensed tannins have a critical influence on root decomposition (**Fig. R2, Table R4**).

Indeed, we agree that incorporating other chemical traits will help us gain a more comprehensive understanding of the afterlife effects of root chemical defense on root decomposition. Therefore, we have added a discussion addressing this point in the main text (Lines 361-365).

“In addition to condensed tannins, lignin, bound phenolics, and other non-structural secondary compounds are also important chemical traits driving root decomposition (Sun et al., 2018; Xia et al., 2021). Further incorporating these chemical drivers would thus help give a more comprehensive picture of the afterlife effects of root defense within the ADD framework.”

Fig. R2 | Coefficients of Pearson's correlation among ML, CTs, RN and C/N for (a) all 90 tree species; (b) 65 arbuscular mycorrhizal (AM) species; (c) 25 ectomycorrhizal (EcM) species. ML, mass loss (decomposition); CTs, condensed tannins, RN, root nitrogen concentration; C/N, root C/N.

Table R4 | Coefficients of Partial correlation (based on Pearson's method) among ML, CTs, RN and C/N.

	Mass loss (response variable)			
	Variables	Controlled variables	Coefficient r	p
All species $n = 90$	CTs	—	-0.464	<0.001
		RN	-0.398	<0.001
		C/N	-0.373	<0.001
		RN + C/N	-0.376	<0.001
	RN	—	0.293	0.005
		CTs	0.144	0.18
	C/N	—	-0.323	0.002
		CTs	-0.133	0.213
AM species $n = 65$	CTs	—	-0.547	<0.001
		RN	-0.499	<0.001
		C/N	-0.482	<0.001
		RN+C/N	-0.483	<0.001
	RN	—	0.264	0.033
		CTs	0.056	0.659
	C/N	—	-0.297	0.016
		CTs	-0.043	0.738
EcM species $n = 25$	CTs	—	-0.680	<0.001
		RN	-0.678	<0.001
		C/N	-0.653	<0.001
		RN + C/N	-0.621	0.002
	RN	—	0.106	0.613
		CTs	-0.072	0.74
	C/N	—	-0.253	0.222
		CTs	0.013	0.951

Sixth, since the authors have examined root anatomy, they can incorporate the data on mycorrhizal colonization in this study as the symbiont dimension, the key trait dimension in this study, describe the mycorrhizal association. As for EM roots, I suggest the authors could give the exact picture or data on how the ECM fungi affect on the cell walls of the roots. This is a direct evidence for the authors' argument of their hypothesis regarding the sharp difference between AM and EM roots in their root- mycorrhizae interaction.

Response: Thanks for the thoughtful comments. It was in our plan to measure mycorrhizal colonization intensity (MCI), but a substantial part of the data was accidentally lost during the campaign and the few species MCI data available are poorly representative of the full species dataset. Considering that the colonization rate is largely influenced by seasonal variation and sampling location/soil microenvironment (Brundrett, 1991; Heinemeyer et al., 2004), we did not measure this trait in the subsequent sampling and the quantification of anatomic traits. Similar to our previous study (Yan et al., 2022), we used the dominant traits (RD, CT, and BI) in conjunction with the axis scores of 'symbiosis gradient' as a proxy of fungal-symbiosis dependency in root strategy. We have now explicitly pointed out the necessity to further incorporate direct mycorrhizal symbiotic traits (colonization rate, carbon investment, and a series of mycorrhizal traits) within the ADD framework in the future (see Lines 374-377).

"...it is still necessary to further examine how a suite of traits with more direct links with the intensity of symbiotic associations (e.g. colonization rates and mycorrhizal C investments) fit within the ADD framework, particularly over long-time scales."

The ADD hypothesis we proposed, including the root-fungal symbiosis in EcM angiosperms species, was built upon previous literature (Weiss et al., 1997; Feugey et al., 1999; Jones et al., 2001; Ragnelli et al., 2014). In the current study, we aimed to provide some support for this framework by examining relationships among root traits, without validating the mechanisms of root-fungal interactions at the microscopic level (e.g., physiological and cellular levels), because we did not use more precise instruments/equipment (e.g. digital slice scanner in conjunction with a transmission electron microscope) when initially quantifying cortical cells. After consulting with professional researchers and technicians, we believe that the resolution of existing root anatomy images (obtained by the traditional photographed using a compound microscope method; **Fig. R4**) is insufficient to identify the root and fungal structures at the microscopic level (e.g. the thickness of the cell wall of root exodermis). We agree that the point you raised is important because previous studies on root-fungal interactions at the microscopic level (e.g. molecular and cell biology) are mostly based on a few species. Therefore, further tests on more species are still needed, and we have emphasized this point in the Discussion (Lines 349-351).

"Nonetheless, further investigation is needed to examine how symbiosis and defense interact to affect the root cell wall at the microscale."

(a) *Castanopsis fargesii*

(b) *Castanopsis fordii*

Fig. R4 | Typical anatomical structures of absorptive roots in *Castanopsis fargesii* (a) and *Castanopsis fordii* (b).

References

- Bergmann, J. et al. The fungal collaboration gradient dominates the root economics space in plants. *Science Advances* **6**, eaba3756 (2020).
- Brundrett, M. Mycorrhizas in natural ecosystems. *Advances in Ecological Research* **21**, 171-313 (1991).
- Eissenstat, D. M. & Volder, A. The efficiency of nutrient acquisition over the life of a root. *Nutrient Acquisition by Plants: Ecological Perspective* **181**, 185-220 (2004).
- Feugey, L., Strullu, D. G., Poupard, P. & Simoneau, P. Induced defence responses limit Hartig net formation in ectomycorrhizal birch roots. *New Phytologist* **144**, 541-547 (1999).
- Heinemeyer, A. et al. Impact of soil warming and shading on colonization and community structure of arbuscular mycorrhizal fungi in roots of a native grassland community. *Global Change Biology* **10**, 52-64 (2004).
- Hobbie, S. E., Oleksyn, J., Eissenstat, D. M. & Reich, P. B. Fine root decomposition rates do not mirror those of leaf litter among temperate tree species. *Oecologia* **162**, 505-513 (2010).
- Huys, R. et al. Plant litter chemistry controls coarse-textured soil carbon dynamics. *Journal of Ecology* **110**, 2911-2928 (2022).
- Jiang, L. et al. Mycorrhizal and environmental controls over root trait-decomposition linkage of woody trees. *New Phytologist* **229**, 284-295 (2021).
- Jiang, L., Kou, L. & Li, S. Alterations of early-stage decomposition of leaves and absorptive roots by deposition of nitrogen and phosphorus have contrasting mechanisms. *Soil Biology Biochemistry* **127**, 213-222 (2018).
- Jones, A. M. Programmed cell death in development and defense. *Plant Physiology* **125**, 94-97

- (2001).
- Ma, C., Xiong, Y., Li, L. & Guo, D. Root and leaf decomposition become decoupled overtime: implications for below- and above-ground relationships. *Functional Ecology* **30**, 1239-1246 (2015).
- Ragnelli, A. M. et al. The cell death phenomenon during Tuber ectomycorrhiza morphogenesis. *Plant Biosystems* **148**, 473-482 (2014).
- Resendes, M. L., Bryla, D. R. & Eissenstat, D. M. Early events in the life of apple roots: variation in root growth rate is linked to mycorrhizal and nonmycorrhizal fungal colonization. *Plant and Soil* **313**, 175-186 (2008).
- Roumet, C. et al. Root structure-function relationships in 74 species: evidence of a root economics spectrum related to carbon economy. *New Phytologist* **210**, 815-826 (2016).
- Sun, T. et al. Contrasting dynamics and trait controls in first-order root compared with leaf litter decomposition. *Proceedings of the National Academy of Sciences of the United States of America* **115**, 10392-10397 (2018).
- Valverde-Barrantes, O. J., Authier, L., Schimann, H. & Baraloto, C. Root anatomy helps to reconcile observed root trait syndromes in tropical tree species. *American Journal of Botany* **108**, 744-755 (2021).
- Weiss, M. et al. Tissue-specific and development-dependent accumulation of phenylpropanoids in larch mycorrhizas. *Plant Physiology* **114**, 15-27 (1997).
- Xia, M., Valverde-Barrantes, O. J., Suseela, V., Blackwood, C. B. & Tharayil, N. Coordination between compound-specific chemistry and morphology in plant roots aligns with ancestral mycorrhizal association in woody angiosperms. *New Phytologist* **232**, 1259-1271 (2021).
- Yan, H. et al. Mycorrhizal symbiosis pathway and edaphic fertility frame root economics space among tree species. *New Phytologist* **234**, 1639-1653 (2022).
- Zhang, Y. et al. The origin of bi-dimensionality in plant root traits. *Trends in ecology & evolution* **39**, 78-88 (2024).

REVIEWERS' COMMENTS

Reviewer #1 (Remarks to the Author):

Review NCOMMS-23-56952A

A trait-based root acquisition-defence-decomposition framework in angiosperm tree species

Congratulations to the authors on an even better revised manuscript. Thank you for incorporating the suggestions from the previous version of the manuscript. The manuscript is in terrific state and I only have a few points on the revised version.

1. Thank you for reminding me of your previous paper (Yan et al. 2022), which my whole lab group enjoyed when it was first published. I understand the current manuscript has a very different aim and set of research questions that build on the previous publication. However, I was unsure how the field sampling of the two papers coincided, since there is considerable overlap in timing and some of the species (i.e., L148)? Was it one field campaign that was divided for separate studies? Two completely separate field campaigns? Assuming these two studies are related, it would be good to highlight that in the main text. The previous study sets the stage really well for the development of the ADD conceptual framework and can fill in the background for readers without using up valuable word counts. There is explicit mention of the previous article in the main text, but the linkage between the studies/papers could be made clearer.
2. The answer to Point 1 may solve a query I have regarding the statistical approach. If the two studies are indeed related (ie., from the same sampling effort), then the standardized z-scores make perfect sense to me, as the analysis on raw values has previously been reported for many of the traits. However, if this study is completely separate from the previous study, then it would be great to see the statistics on raw values, as this adds a lot to the ecological inference.
3. Fig. 2e. Which family is the grey box that is sat to the lower left of the line? Consider labelling it.
4. Fig. S6. It seems like the outlier family is an AM, consider labelling this one here as well. I was also confused for a moment because the Juglandaceae species I work with are all ECM. I wonder if it is worth making a note on the figure (and throughout) that *Juglans* is AM associated. I'm not sure how many of your readers have similar Juglandaceae bias, but it is definitely worth noting that it is one of the few plant families that have multiple symbiont types.
5. The other Reviewer made some very valid comments on methodologies and the authors' responses were excellent. I want to commend both for a very interesting discussion that I learned a lot from! However, I do want to reinforce the authors' responses as root research continues to be underrepresented in the literature and the methodological difficulties are undervalued compared to sampling other plant organs.
 - 1) Sampling roots of 90 species is extraordinary! (The high species richness and taxonomic ranges make the constrained root diameter very interesting!) I can only think of 1 study that including more species (Weemstra et al. 2023). Plus, as the authors mention these 90 species represent 4 sites and 2 biomes—a commendable sampling effort!
 - 2) Similarly, there are very few studies that have quantified root decomposition rates and, as the authors acknowledge, using live roots is currently the only method we have for controlled experiments.

- 3) It is also worth noting that controlled incubation studies is a valid method for comparing species k – values or mass loss over a specific time period, for both roots and leaves of species, and the length of the experiment should only be relevant insofar that it is the same for all samples (ie., an experimental decomposition value that is comparable across species and sites, not the absolute value is the interest here).
- 4) Great point on the RD range. These species have similar RD and variation compared to our compiled species-specific datasets for [TropiRoots](https://tropiroottrait.github.io/TropiRootTrait/index.html) (<https://tropiroottrait.github.io/TropiRootTrait/index.html>) and GRooT datasets, which includes temperate, subtropical and tropical species (~3000 species)—and actually includes a higher max RD value compared to the global dataset! The focus on 1st and 2nd order roots – as compared to < 2mm cutoff that many researchers use – may well explain the low RD and variation in both cases. Further sampling would be unlikely to add much range unless root orders or plant types were expanded (ie., lianas or subtropical herbs/palms).
- 5) Thank you to the Reviewer and authors on the insightful discussion on non-structural vs structural defence chemistry (ie., CTs vs. lignin). I am glad to see this make it to the main text!
- 6) I also enjoyed the discussion regarding mycorrhizal colonization. Again the authors made some excellent points and I am glad to see the discussion contributing to the revised text. I was wondering if the root anatomy figures (Fig. R4) could be incorporated into the supplementary material? They are beautiful anatomical scans that rarely make it into root trait ecology papers.

Wonderful job on the revisions and a delightful manuscript to review!

Cheers,

Dr. Kelly M. Andersen

Reviewer #2 (Remarks to the Author):

A thorough explanation has already been provided for the questions I raised, but I still have some issues I would like to feedback to the author. When the author responded to my fourth question, I believe that even for woody plants, the variation in root diameter can be quite significant. This point, however, was not addressed to my satisfaction by the author. If possible, I hope the author can provide a more reasonable explanation, or at least acknowledge this limitation (the gradient of root diameters in the article is relatively small) within the article, and I hope that future experiments can offer a more comprehensive explanation in this regard.

Point-by-point responses to reviewers' comments

Reviewer #1 (Remarks to the Author):

Congratulations to the authors on an even better revised manuscript. Thank you for incorporating the suggestions from the previous version of the manuscript. The manuscript is in terrific state and I only have a few points on the revised version.

1. Thank you for reminding me of your previous paper (Yan et al. 2022), which my whole lab group enjoyed when it was first published. I understand the current manuscript has a very different aim and set of research questions that build on the previous publication. However, I was unsure how the field sampling of the two papers coincided, since there is considerable overlap in timing and some of the species (i.e., L148)? Was it one field campaign that was divided for separate studies? Two completely separate field campaigns? Assuming these two studies are related, it would be good to highlight that in the main text. The previous study sets the stage really well for the development of the ADD conceptual framework and can fill in the background for readers without using up valuable word counts. There is explicit mention of the previous article in the main text, but the linkage between the studies/papers could be made clearer.

2. The answer to Point 1 may solve a query I have regarding the statistical approach. If the two studies are indeed related (ie., from the same sampling effort), then the standardized z-scores make perfect sense to me, as the analysis on raw values has previously been reported for many of the traits. However, if this study is completely separate from the previous study, then it would be great to see the statistics on raw values, as this adds a lot to the ecological inference.

Response: We really appreciate your acknowledgment of our previous work on root economics space. Indeed, this was one field campaign that provided material for two separate studies. Following your suggestion, we have mentioned the previous study in the main text and clarified the relationship between these two different studies (Lines 145-148).

“In this study, we focused on absorptive roots (the most distal 1st- and 2nd-order roots within the branching root system) of 90 angiosperm tree species (65 AM and 25 EcM) spanning two temperate and two subtropical forests in China. These species represent a subset of species from Yan et al.¹⁸.”

3. Fig. 2e. Which family is the grey box that is sat to the lower left of the line? Consider labelling it.

Response: The gray box at the bottom left is the Rosaceae family. Following your suggestion, we have labelled it in the Fig. 2e.

4. Fig. S6. It seems like the outlier family is an AM, consider labelling this one here as well. I was also confused for a moment because the Juglandaceae species I work with are all ECM. I wonder if it is worth making a note on the figure (and throughout) that Juglans is AM associated. I'm not sure how many of your readers have similar Juglandaceae bias, but it is definitely worth noting that it is one of the few plant families that have multiple symbiont types.

Response: This family is still the Rosaceae, and we have labelled it in the Fig. S6. As you pointed out, Juglandaceae is indeed a plant family with multiple symbiotic types. Our study involves three species from the Juglandaceae. one (*Juglans cathayensis*) is AM and the other two (*Juglans regia*, *Juglans mandshurica*) are AM/EcM (facultatively) based on the Plant species mycorrhizal trait dataset (Meng et al., 2023). These three species are all AM based on our observations. The mycorrhizal status of certain plants may vary with soil condition and plant ontogenetic stage, leading to discrepancies in the dominant mycorrhizal types among studies. In this analysis, we first selected all AM species and then retained families that included at least three species. For clarity, we have added a note to highlight this point in the legend of Fig. S6.

“Notes: The families involved here refer to those to which the AM species in this study belong, and does not mean that all species in this family are AM.”

References:

Meng, Y., Davison, J., Clarke, J.T., Zobel, M., Gerz, M., Moora, M. et al. (2023) Environmental modulation of plant mycorrhizal traits in the global flora. *Ecology Letters*, 26, 1862–1876.

<https://doi.org/10.1111/ele.14309>

5. The other Reviewer made some very valid comments on methodologies and the authors' responses were excellent. I want to commend both for a very interesting discussion that I learned a lot from! However, I do want to reinforce the authors' responses as root research continues to be underrepresented in the literature and the methodological difficulties are undervalued compared to sampling other plant organs.

1) Sampling roots of 90 species is extraordinary! (The high species richness and taxonomic ranges make the constrained root diameter very interesting!) I can only think of 1 study that including more species (Weemstra et al. 2023). Plus, as the authors mention these 90 species represent 4 sites and 2 biomes—a commendable sampling effort!

2) Similarly, there are very few studies that have quantified root decomposition rates and, as the authors acknowledge, using live roots is currently the only method we have for controlled experiments.

3) It is also worth noting that controlled incubation studies is a valid method for comparing species k – values or mass loss over a specific time period, for both roots and leaves of species, and the length of the experiment should only be relevant insofar that it is the same for all samples (ie., an experimental decomposition value that is comparable across species and sites, not the absolute value is the interest here).

4) Great point on the RD range. These species have similar RD and variation compared to our compiled species-specific datasets for TropiRoots

(<https://tropiroottrait.github.io/TropiRootTrait/index.html>) and GRooT datasets, which includes temperate, subtropical and tropical species (~3000 species)—and actually includes a higher max RD value compared to the global dataset! The focus on 1st and 2nd order roots – as compared to < 2mm cutoff that many researchers use – may well explain the low RD and variation in both cases. Further sampling would be unlikely to add much range unless root orders or plant types were expanded (ie., lianas or subtropical herbs/palms).

5) Thank you to the Reviewer and authors on the insightful discussion on non-structural vs structural defence chemistry (ie., CTs vs. lignin). I am glad to see this make it to the main text!

6) I also enjoyed the discussion regarding mycorrhizal colonization. Again the authors made some excellent points and I am glad to see the discussion contributing to the revised text. I was wondering if the root anatomy figures (Fig. R4) could be incorporated into the supplementary material? They are beautiful anatomical scans that rarely make it into root trait ecology papers.

Response: We are grateful to you for recognizing our study and manuscript revision, and the valuable suggestions you provided were really helpful in improving our manuscript. As for the anatomy figure, given that all supplementary information needs to be properly cited in the main text, we therefore did not include it in the supplementary material, but it can be seen through this upcoming peer-reviewed document.

Wonderful job on the revisions and a delightful manuscript to review!

Response: Thanks again for your positive feedback and insightful comments on our work!

Cheers,

Dr. Kelly M. Andersen

Reviewer #2 (Remarks to the Author):

A thorough explanation has already been provided for the questions I raised, but I still have some issues I would like to feedback to the author. When the author responded to my fourth question, I believe that even for woody plants, the variation in root diameter can be quite significant. This point, however, was not addressed to my satisfaction by the author. If possible, I hope the author can provide a more reasonable explanation, or at least acknowledge this limitation (the gradient of root diameters in the article is relatively small) within the article, and I hope that future experiments can offer a more comprehensive explanation in this regard.

Response: Thank you so much for your valuable comments. Following your suggestion, we have acknowledged this limitation in the main text (Lines 290-292).

“Nonetheless, we also acknowledge that the low variation in RD observed in our study may not be representative of global tree root variation, which deserves further research.”